# Subcortical serotonin 5HT$_{2c}$ receptor-containing neurons sex-specifically regulate binge-like alcohol consumption, social, and arousal behaviors in mice

M. E. Flanigan[1], O. J. Hon[1,2], S. D'Ambrosio[1], K. M. Boyt[1], L. Hassanein[1], M. Castle[1], H. L. Haun[1], M. M. Pina[1] & T. L. Kash [1,3] ✉

Binge alcohol consumption induces discrete social and arousal disturbances in human populations that promote increased drinking and accelerate the progression of Alcohol Use Disorder. Here, we show in a mouse model that binge alcohol consumption disrupts social recognition in females and potentiates sensorimotor arousal in males. These negative behavioral outcomes were associated with sex-specific adaptations in serotonergic signaling systems within the lateral habenula (LHb) and the bed nucleus of the stria terminalis (BNST), particularly those related to the receptor 5HT$_{2c}$. While both BNST and LHb neurons expressing this receptor display potentiated activation following binge alcohol consumption, the primary causal mechanism underlying the effects of alcohol on social and arousal behaviors appears to be excessive activation of LHb$_{5HT2c}$ neurons. These findings may have valuable implications for the development of sex-specific treatments for mood and alcohol use disorders targeting the brain's serotonin system.

Binge alcohol drinking is a significant cause of alcohol-related death, illness, and economic burden[1]. Moreover, repeated cycles of binge drinking and withdrawal increase the incidence of negative social and emotional states, which can promote further increases in drinking and the transition to alcohol dependence[2]. Thus, understanding the neurobiological mechanisms mediating the negative effects of alcohol consumption on behavior may be critical for limiting subsequent escalations in alcohol intake. In humans, abstinence from binge alcohol consumption is strongly associated with both impaired social emotion recognition behavior and enhanced arousal[3–6], and repeated cycles of binge drinking and abstinence exacerbate these issues[7–13]. However, whether binge alcohol consumption dysregulates social recognition and arousal behavior in adult rodents has not been explored.

The brain's serotonin (5-hydroxytryptamine, 5-HT) system is a critical modulator of affect, motivation, social behavior, arousal, and metabolism[14–17]. Dysregulation of 5-HT signaling has been implicated in the pathophysiology of numerous psychiatric disorders, including major depression, anxiety disorders, substance use disorders, obsessive-compulsive disorder, and schizophrenia[18]. 5-HT neurons are located primarily in hindbrain raphe nuclei, specifically the dorsal (DRN) and median (MRN) raphe nuclei. These neurons project widely throughout the brain, including to cortical, amygdalar, midbrain, and other hindbrain regions (for review, see ref. [19]). A recent anatomical and functional mapping study of the DRN revealed that there are distinct functional populations of 5-HT neurons localized to discrete subregions of the DRN[20]. While 5-HT is implicated in Alcohol Use Disorder (AUD) pathophysiology[21], studies in human and animal models report that manipulations of 5-HT signaling can both increase and decrease alcohol consumption and associated negative affect[21–27]. The heterogeneity of these observed effects is likely due to differential

[1]Bowles Center for Alcohol Studies, University of North Carolina School of Medicine, Chapel Hill, NC, USA. [2]Curriculum in Neuroscience, University of North Carolina School of Medicine, Chapel Hill, NC, USA. [3]Department of Pharmacology, University of North Carolina School of Medicine, Chapel Hill, NC, USA. ✉e-mail: Thomas_kash@med.unc.edu

recruitment of discrete 5-HT circuits and receptor sub-types in AUD regulating specific motivational, cognitive, and social processes.

Both the lateral habenula (LHb) and the bed nucleus of the stria terminalis (BNST) receive inputs from serotonergic neurons located in the dorsal and caudal regions of the dorsal-raphe nucleus[19,28,29] and, according to the aforementioned study[20], may represent two key target regions of an "aversive" subcortical stream of serotonin signaling. Activation of LHb or BNST neurons generally promotes negative emotional states and can reduce motivation for natural and drug rewards[30–35], and large subsets of neurons in both regions are depolarized by 5-HT through activation of the 5HT$_{2c}$ receptor[36–38]. 5HT$_{2c}$ is a primarily Gq-protein-coupled receptor and is highly edited at the RNA level, resulting in a large number of unique isoforms that engage discrete intracellular signaling pathways[39]. Systemic antagonism of 5HT$_{2c}$ has been shown to reduce behaviors related to anxiety and depression in both alcohol-exposed and alcohol-naïve male rodents[25,40,41], but both 5HT$_{2c}$ agonists and antagonists can reduce alcohol intake[26,42,43]. While few studies have investigated the effects of functional modulation of LHb or BNST 5HT$_{2c}$ receptors on behavior, antagonism of 5HT$_{2c}$ in the LHb of male rats reduces both alcohol self-administration and withdrawal-induced anxiety-like behavior in the open field test[24]. These studies support the potential role of 5HT$_{2c}$ signaling in these structures in the regulation of alcohol drinking and affective behavior while also highlighting the need for future studies, particularly in female subjects.

In this study, we first characterized the anatomy, physiology, and behavioral function of the DRN-LHb and DRN-BNST projections in binge alcohol drinking, arousal behavior, and social behavior, with a specific focus on 5-HT inputs to 5HT$_{2c}$-receptor-containing neurons in the LHb and the BNST. We establish a model whereby three weeks of binge alcohol intake induces long-lasting behavioral and physiological adaptations that are similar to those observed in binge-drinking humans. Using complementary approaches for brain region-specific genetic deletion of 5HT$_{2c}$ expression and chemogenetic manipulation of G-protein signaling in 5HT$_{2c}$ neurons, we identify sex- and region-specific roles for 5HT$_{2c}$ receptors themselves as well as 5HT$_{2c}$-containing neurons in social and arousal behaviors in the context of binge alcohol consumption. Overall, our results suggest that binge alcohol consumption potentiates DRN-BNST and DRN-LHb circuitry to promote disruptions in affective behavior, but that excessive activation of the LHb$_{5HT2c}$ neurons is the primary causal mechanism promoting behavioral dysfunction by alcohol.

## Results

### LHb$_{5HT2c}$ and BNST$_{5HT2c}$ neurons mount coordinated responses to rewarding and aversive stimuli

Given the potential role of 5HT$_{2c}$ signaling in the LHb and BNST in regulating binge drinking and affective behavior, we first wanted to understand how neurons that contain these receptors respond to aversive and rewarding stimuli in-vivo. First, male and female 5HT$_{2c}$-cre mice[44] were unilaterally injected with a cre-dependent AAV encoding the calcium sensor GCaMP7f in the LHb and BNST and optical fibers were implanted above these regions. Using fiber photometry we recorded calcium signals from both LHb$_{5HT2c}$ and BNST$_{5H2c}$ neurons during free social interaction with a novel juvenile same-sex conspecific, the acoustic startle test, voluntary water drinking, and voluntary alcohol drinking (Fig. 1A, B). We focused our efforts on these behaviors as they have been previously linked to 5HT$_{2c}$ function in these regions and are disrupted in human binge drinkers. Importantly, we were able to correct for any motion artifacts during recording by subtracting out signal from our isosbestic control wavelength channel (415 nm). In general, both populations of 5HT$_{2c}$ neurons in both sexes were activated by social interaction and acoustic startle stimuli, but inhibited by water and alcohol consumption (Fig. 1C-T). However, males displayed more robust BNST$_{5HT2c}$ responses to each of these stimuli. For example, relative to females, males displayed increased

activation in response to social targets and acoustic startle stimuli (Fig. 1M-P), but stronger inhibition in response to alcohol (Fig. 1Q, R) and water consumption (Fig. 1S, T). In LHb$_{5HT2c}$ neurons, however, females mounted a response to water consumption (Fig. 1I, J) but not to alcohol consumption (Fig. 1K, L), while males mounted similarly strong responses to both water (Fig. 1S, T) and alcohol (Fig. 1Q, R). Notably, mice were water-deprived prior to voluntary drinking experiments in order to facilitate consumption while tethered to the fiber photometry apparatus. Under these conditions, we would expect that both water and alcohol consumption would be rewarding, yet only water consumption elicited a response in females, which may be suggestive of mixed 'reward/aversion' effects of alcohol in binge-naïve LHb. Remarkably, LHb$_{5HT2c}$ and BNST$_{5HT2c}$ neurons displayed robust overlap of their activity during these behaviors (Fig. 1C, D), suggesting that their activity may be regulated by common upstream inputs.

To characterize 5-HT dynamics, a separate cohort of 5HT$_{2c}$-cre male and female mice were unilaterally injected with a cre-dependent AAV encoding the 5-HT sensor GRAB-5HT in the LHb and BNST and optical fibers were implanted above these regions (Fig. 2A, B). This sensor fluoresces upon binding of 5-HT, which allows us to visualize 5-HT dynamics with millisecond temporal resolution[45]. We determined that motion artifacts were not occurring during these recordings by performing recordings in a separate cohort of mice using a cre-dependent GFP control virus (Fig. S7). This is necessary because the isosbestic control wavelength for this sensor has not yet been verified. Similar to the calcium responses observed in LHb$_{5HT2c}$ and BNST$_{5HT2c}$ neurons, 5-HT release onto these neurons was generally increased in response to social interaction and acoustic startle stimuli but decreased during water and alcohol consumption (Fig. 2E–T). BNST$_{5HT2c}$ and LHb$_{5HT2c}$ neurons also displayed robust overlap of 5-HT signals during these behaviors (Fig. 2C, D). Interestingly, females showed stronger reductions in BNST$_{5HT2c}$ 5-HT during water and alcohol consumption compared to males (Fig. 2Q–T). Females also displayed faster decay of LHb$_{5HT2c}$ and BNST$_{5HT2c}$ 5-HT signals following acoustic startle, as evidenced by a reduced time constant (tau) when fit with a one-phase decay exponential curve (Fig. 2G, H, O–P). This longer duration of 5-HT signaling could be a potential mechanism mediating more robust modulation of 5HT2c-containing neurons in response to valenced stimuli in males compared to females (as observed in Fig. 1M–T).

To investigate the anatomy and neurochemical identity of the DRN-LHb and DRN-BNST projections, we next performed dual region retrograde tracing combined with immunohistochemistry for 5-HT (Fig. 3A–D). In both males and females, a vast majority of DRN neurons projecting to the LHb and BNST were positive for 5-HT (Fig. 3M, N). In addition, we observed that an extremely high percentage of DRN neurons project to both BNST and LHb (Fig. 3O). We hesitated to directly compare percentages of these overlapping neurons between males and females, as differences in infection spread and uptake are challenging to normalize in these experiments. However, it may be the case that males have fewer DRN-BNST neurons that also project to the LHb than females (Fig. 3P), and this should be investigated further in future studies using more quantitative tracing methods. Consistent with previous reports[20,46], the anatomical localization of these BNST- and LHb-projecting 5-HT neurons in the DRN was largely in dorsal and caudal regions. Subsequent in-situ hybridization experiments revealed that BNST$_{5HT32c}$ neurons are primarily co-localized with the GABAergic marker vGAT, while LHb$_{5HT32c}$ neurons are primarily co-localized with the glutamatergic marker vGlut3 (Fig. 3Q–T). Notably, the percentages of BNST$_{5HT32c}$/vGAT and LHb$_{5HT32c}$/vGlut2 were comparable between sexes (Fig. 3U–W).

### Gq signaling in BNST$_{5HT2c}$ and LHb$_{5HT2c}$ modulates social behavior, arousal behavior, and binge-like alcohol consumption

Given that the 5HT$_{2c}$ receptor primarily signals through Gq-coupled signaling mechanisms[47], we next asked how activation of Gq signaling

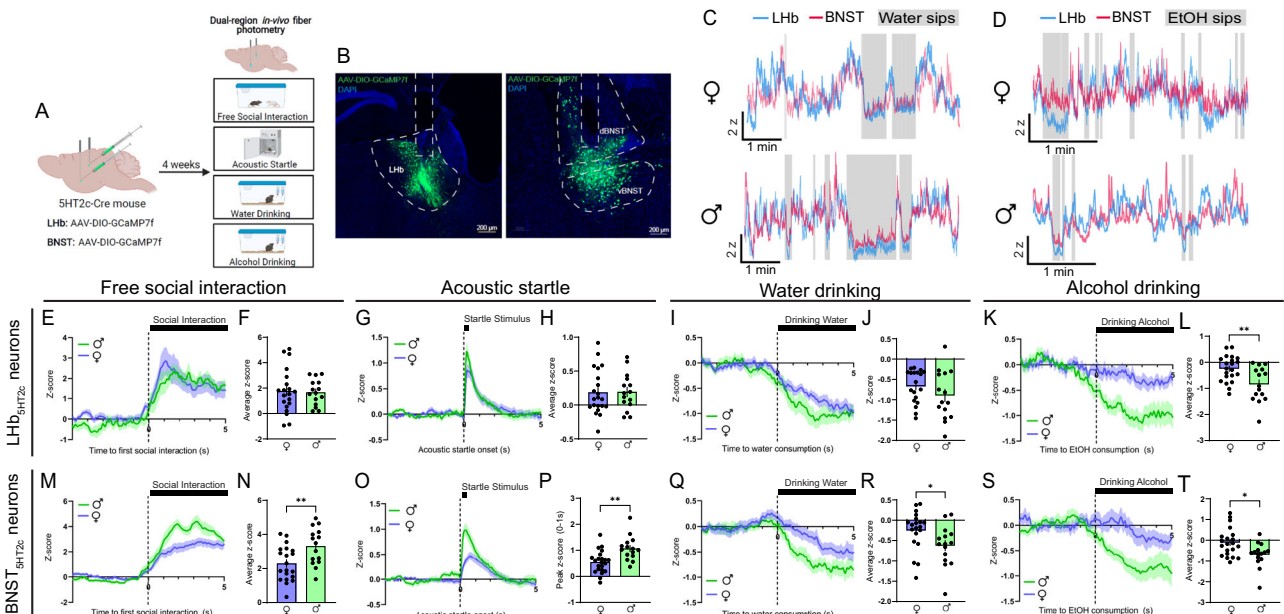

**Fig. 1 | In-vivo LHb_{5HT2c} and BNST_{5HT2c} calcium signals are modulated by exposure to affective stimuli. A**, Surgical schematic and experimental timeline for GCaMP fiber photometry experiments. **B** Representative viral infections and fiber placements in LHb (left) and BNST (right). Similar viral placement was reproduced in $n = 37$ mice. **C** Representative traces for water drinking session (individual mice). **D** Representative traces for alcohol drinking session (individual mice). **E** Peri-event plot of LHb_{5HT2c} GCaMP activity during the first interaction with a novel juvenile social target (Females $n = 20$ mice/1 trial; Males $n = 15$ mice/1 trial). **F** Average z-score of LHb_{5HT2c} GCaMP signal for 0-5 s post interaction (Females $n = 20$ mice/1 trial, Males: $n = 15$ mice/1 trial). **G** Peri-event plot of LHb_{5HT2c} GCaMP activity during the acoustic startle test (Females: $n = 21$ mice/10 trials; Males: $n = 15$ mice/10 trials). **H** Average z-score of LHb_{5HT2c} GCaMP signal for 0-5 s post startle stimulus (Females: $n = 21$ mice/10 trials; Males: $n = 15$ mice/10 trials). **I** Peri-event plot of LHb_{5HT2c} GCaMP activity during voluntary water consumption (Females: $n = 21$ mice/1-4 bouts; Males: $n = 15$ mice/1-4 bouts). **J** Average z-score of LHb_{5HT2c} GCaMP signal for 0-5 s post water drinking bout start (Females: $n = 21$ mice/1-4 bouts; Males: $n = 15$ mice/1-4 bouts). **K** Peri-event plot of LHb_{5HT2c} GCaMP activity during voluntary alcohol consumption (Females: $n = 21$ mice/1-4 bouts; Males: $n = 15$ mice/1-4 bouts). **L** Average z-score of LHb_{5HT2c} GCaMP signal for 0-5 s post alcohol drinking bout start, $p = 0.0038$ (Females: $n = 21$ mice/1-4 bouts; Males: $n = 15$ mice/

1-4 bouts). **M** Peri-event plot of BNST_{5HT2c} GCaMP activity during the first inter-action with a novel juvenile social target (Females: $n = 20$ mice/1 trial; Males: $n = 15$ mice/1 trial). **N** Average z-score of BNST_{5HT2c} GCaMP signal for 0-5 s post interac-tion, $p = 0.0073$ (Females: $n = 20$ mice/1 trial; Males: $n = 15$ mice/1 trial). **O** Peri-event plot of BNST_{5HT2c} GCaMP activity during the acoustic startle test (Females: $n = 22$ mice/10 trials; Males: $n = 15$ mice/10 trials). **P** Peak z-score of BNST_{5HT2c} GCaMP signal for 0-1 s post startle stimulus, $p = 0.0012$ (Females: $n = 22$ mice/10 trials; Males: $n = 15$ mice/10 trials). **Q** Peri-event plot of BNST_{5HT2c} GCaMP activity during voluntary water consumption (Females: $n = 20$ mice/1-4 bouts; Males: $n = 15$ mice/1-4 bouts). **R** Average z-score of BNST_{5HT2c} GCaMP signal for 0-5 s post water drinking bout start, $p = 0.0296$ (Females: $n = 20$ mice/1-4 bouts; Males: $n = 15$ mice/1-4 bouts). **S** Peri-event plot of BNST_{5HT2c} GCaMP activity during voluntary alcohol consumption (Females: $n = 21$ mice/1-4 bouts; Males: $n = 15$ mice/1-4 bouts). **T** Average z-score of BNST_{5HT2c} GCaMP signal for 0-5 s post water drinking bout start, $p = 0.0103$ (Females: $n = 21$ mice/1-4 bouts; Males: $n = 15$ mice/1-4 bouts). Unless otherwise stated, statistical comparisons were performed using a two-tailed *Student's* unpaired *t* test. All data are represented as mean ± SEM. *$p < 0.05$, **$p < 0.01$. Source data are provided as a Source Data file. Created with Biorender.com.

in LHb_{5HT2c} and BNST_{5HT2c} would impact binge drinking, social, and arousal behavior in alcohol-naïve mice. 5HT_{2c}-cre mice were injected in the LHb or BNST with a cre-dependent AAV encoding either the Gq-coupled DREADD hM3Dq or a mCherry control virus (Figs. 4A, K, S14) and three weeks later were tested in the 3-chamber social test, acoustic startle test, open field test, and sucrose preference test before being tested in a binge alcohol consumption test. To evaluate binge alcohol consumption, we employed the widely used Drinking in the Dark (DiD) model. In the DiD paradigm, animals are given free access to 20% alcohol in water for two hours on Monday–Wednesday and four hours on Thursday and this pattern is repeated for 3 weeks (cycles). While female mice generally consume greater amounts of alcohol than male mice in this model, both sexes voluntarily drink to achieve intoxication at blood alcohol concentrations (BACs) of at least 80 mg/dl[48]. An acute intraperitoneal dose of 3 mg/kg clozapine-n-oxide (CNO) was admi-nistered to all mice 30 min prior to each individual behavioral task, while CNO treatment for the binge drinking test occurred only on the last day of DiD. We verified this CNO treatment significantly increased expression of the activity-dependent marker cFos in DREADD-treated animals (Fig. S1). Critically, aside from a small effect on alcohol pre-ference, CNO treatment alone in the absence of viral expression did not impact the behaviors tested (Fig. S2). Activation of Gq signaling in

BNST_{5HT2c} neurons did not affect social behavior in the 3-chamber social preference or social recognition tasks in males, but an interac-tion between sex and virus treatment was observed for social pre-ference such that females displayed lower social preference in DREADD-treated animals compared water treated animals and males displayed higher social preference in DREADD-treated animals com-pared to control animals (Fig. 4C, D). Activation of Gq signaling in BNST_{5HT2c} neurons increased acoustic startle responses in both sexes (Fig. 4E–G) but reduced sucrose and alcohol drinking only in females (Fig. 4H–J). DREADD-mediated activation of BNST_{5HT2c} neurons had no effect on open-field exploratory or locomotor behavior in either sex (Fig. S3). These findings indicate an important sex difference in the behavioral role of Gq signaling in BNST_{5HT2c} neurons: that it primarily promotes arousal in males, whereas it promotes arousal and reduces consumption of rewards in females.

While DREADD-mediated activation of LHb_{5HT2c} neurons pri-marily altered behaviors in a similar fashion to DREADD-mediated activation of BNST_{5HT2c} neurons, sex differences in these behavioral effects were not observed. Indeed, activation of LHb_{5HT2c} neurons reduced social preference, sucrose consumption, and alcohol con-sumption in both sexes (Fig. 4K–T). However, contrary to what we observed in the BNST, activation of LHb_{5HT2c} neurons reduced

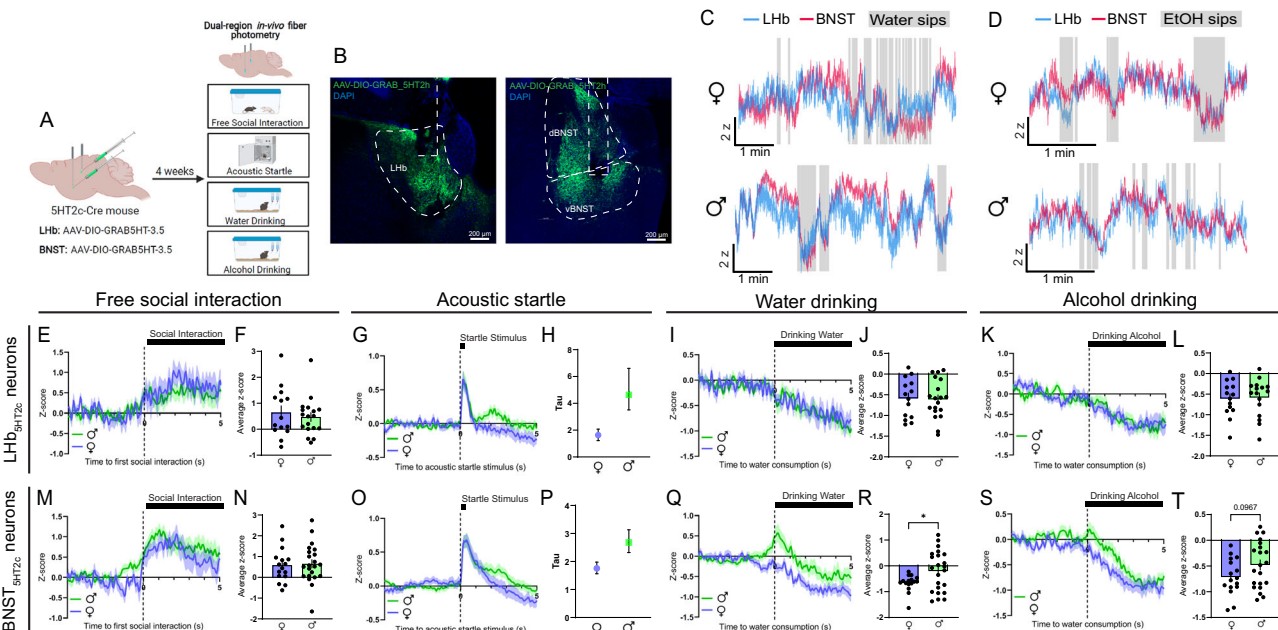

**Fig. 2 | In-vivo LHb$_{5HT2c}$ and BNST$_{5HT2c}$ serotonin signals are modulated by exposure to affective stimuli. A** Surgical schematic and experimental timeline for GRAB-5HT fiber photometry experiments. **B** Representative viral infections and fiber placements in LHb (left) and BNST (right). Similar viral placement was reproduced in $n = 39$ mice. **C**, Representative traces for water drinking session. **D** Representative traces for alcohol drinking session. **E** Peri-event plot of LHB$_{5HT2c}$ GRAB-5HT activity during the first interaction with a novel juvenile social target (Females: $n = 14$ mice/1 trial; Males: $n = 18$ mice/1 trial). **F** Average z-score of LHb$_{5HT2c}$ GRAB-5HT signal for 0-5 s post interaction (Females: $n = 14$ mice/1 trial; Males: $n = 18$ mice/1 trial). **G** Peri-event plot of LHb$_{5HT2c}$ GRAB-5HT activity during the acoustic startle test (Females: $n = 14$ mice/10 trials; Males: $n = 18$ mice/10 trials). **H** Tau (time constant) of LHb$_{5HT2c}$ GRAB-5HT signal for 0-5 s post startle stimulus (fit with one phase exponential decay curve, bars depict 95% confidence interval) (Females: $n = 14$ mice/10 trials; Males: $n = 18$ mice/10 trials). **I** Peri-event plot of LHb$_{5HT2c}$ GRAB-5HT activity during voluntary water consumption (Females: $n = 14$ mice/1-4 bouts; Males: $n = 19$ mice/1-4 bouts). **J** Average z-score of LHb$_{5HT2c}$ GRAB-5HT signal for 0-5 s post water drinking bout start (Females: $n = 14$ mice/1-4 bouts; Males: $n = 19$ mice/1-4 bouts). **K** Peri-event plot of LHb$_{5HT2c}$ GRAB-5HT activity during voluntary alcohol consumption (Females: $n = 14$ mice/1-4 bouts; Males: $n = 16$ mice/1-4 bouts). **L** Average z-score of LHb$_{5HT2c}$ GRAB-5HT signal for 0-5 s post alcohol drinking bout start (Females: $n = 14$ mice/1-4 bouts; Males: $n = 16$ mice/

1-4 bouts). **M** Peri-event plot of BNST$_{5HT2c}$ GRAB-5HT activity during the first interaction with a novel juvenile social target (Females: $n = 16$ mice/1 trial; Males: $n = 21$ mice/1 trial). **N** Average z-score of BNST$_{5HT2c}$ GRAB-5HT signal for 0-5 s post interaction (Females: $n = 16$ mice/1 trial; Males: $n = 21$ mice/1 trial). **O** Peri-event plot of BNST$_{5HT2c}$ GRAB-5HT activity during the acoustic startle test (Females: $n = 16$ mice/10 trials; Males: $n = 21$ mice/10 trials). **P** Tau (time constant) of BNST$_{5HT2c}$ GRAB-5HT signal for 0-5 s post startle stimulus (fit with one phase exponential decay curve, bars depict 95% confidence interval) (Females: $n = 16$ mice/10 trials; Males: $n = 21$ mice/10 trials). **Q** Peri-event plot of BNST$_{5HT2c}$ GRAB-5HT activity during voluntary water consumption (Females: $n = 16$ mice/1-4 bouts; Males: $n = 23$ mice/1-4 bouts). **R** Average z-score of BNST$_{5HT2c}$ GRAB-5HT signal for 0-5 s post water drinking bout start, $p = 0.0331$ (Females: $n = 16$ mice/1-4 bouts; Males: $n = 23$ mice/1-4 bouts). **S** Peri-event plot of BNST$_{5HT2c}$ GRAB-5HT activity during voluntary alcohol consumption (Females: $n = 16$ mice/1-4 bouts; Males: $n = 21$ mice/1-4 bouts). **T** Average z-score of BNST$_{5HT2c}$ GRAB-5HT signal for 0-5 s post water drinking bout start (Females: $n = 16$ mice/1-4 bouts; Males: $n = 21$ mice/1-4 bouts). Unless otherwise stated, statistical comparisons were performed using a two-tailed *Student's* unpaired *t* test. All data excluding panels 2H and 2P are represented as mean ± SEM. *$p < 0.05$, **$p < 0.01$. Source data are provided as a Source Data file. Created with Biorender.com.

responses in the acoustic startle test rather than enhanced them. These results are in agreement with an established literature describing a role for the LHb in mediating passive coping to threats[49–51], yet highlight an important divergence in the behavioral functions of Gq signaling in LHb$_{5HT2c}$ versus BNST$_{5HT2c}$ neurons in alcohol-naïve mice. Of further relevance in this context, we observed that chemogenetic activation of LHb$_{5HT2c}$ neurons reduced exploratory behavior, but not locomotion, in an open field (Fig. S3).

**Binge alcohol consumption produces sex-specific affective disturbances and increases 5-HT$_{2c}$ receptor expression in the LHb**
Our results thus far suggest that BNST$_{5HT2c}$ and LHb$_{5HT2c}$ neurons play similar, though in part sex-dependent, roles in modulating social, arousal, and consummatory behaviors in naïve mice. Next, we wanted to ask: how does an insult that disrupts social and arousal behaviors impact the BNST$_{5HT2c}$ and LHb$_{5HT2c}$ systems? Studies in humans and animal models suggest that a history of alcohol consumption reduces social behaviors and increases anxiety-like behaviors, which may involve alcohol-induced adaptations in LHb and/or BNST 5HT$_{2c}$ receptor expression and signaling[22,24,25,27]. To investigate this possibility, we subjected mice to three weeks of DiD, and then

after one week of abstinence assessed their behavior in the 3-chamber sociability task, the open field test, and the acoustic startle test (Fig. 5A). Notably, this behavioral battery was designed in consideration of robust binge drinking-induced dysregulation of facial emotion recognition and acoustic startle responses reported in humans[5,52–55]. Consistent with previous reports, both sexes reached significant levels of intoxication as determined by their BACs at 4 h (Fig. 5B–E). It should be noted that in the DiD paradigm, 4 h BACs may not be fully reflective of peak intoxication, as mice tend to front-load their alcohol consumption, drinking the majority of their total volume in the first two hours[56]. DiD had no impact on social preference, but there was an interaction between sex and DiD such that social recognition was blunted in DiD females but not DiD males (Fig. 5F, G). On the other hand, DiD had no impact on startle in female mice but markedly increased startle in males (Fig. 5L–N). DiD similarly reduced exploratory behavior in both sexes in the open field test, although this effect was not consistent across behavioral cohorts (Fig. 5H–K, also see S8). When we re-tested these mice four weeks following their last DiD exposure, we observed a normalization of female social recognition deficits as well as male and female open-field exploratory deficits (Fig. S4). However, the enhancement of

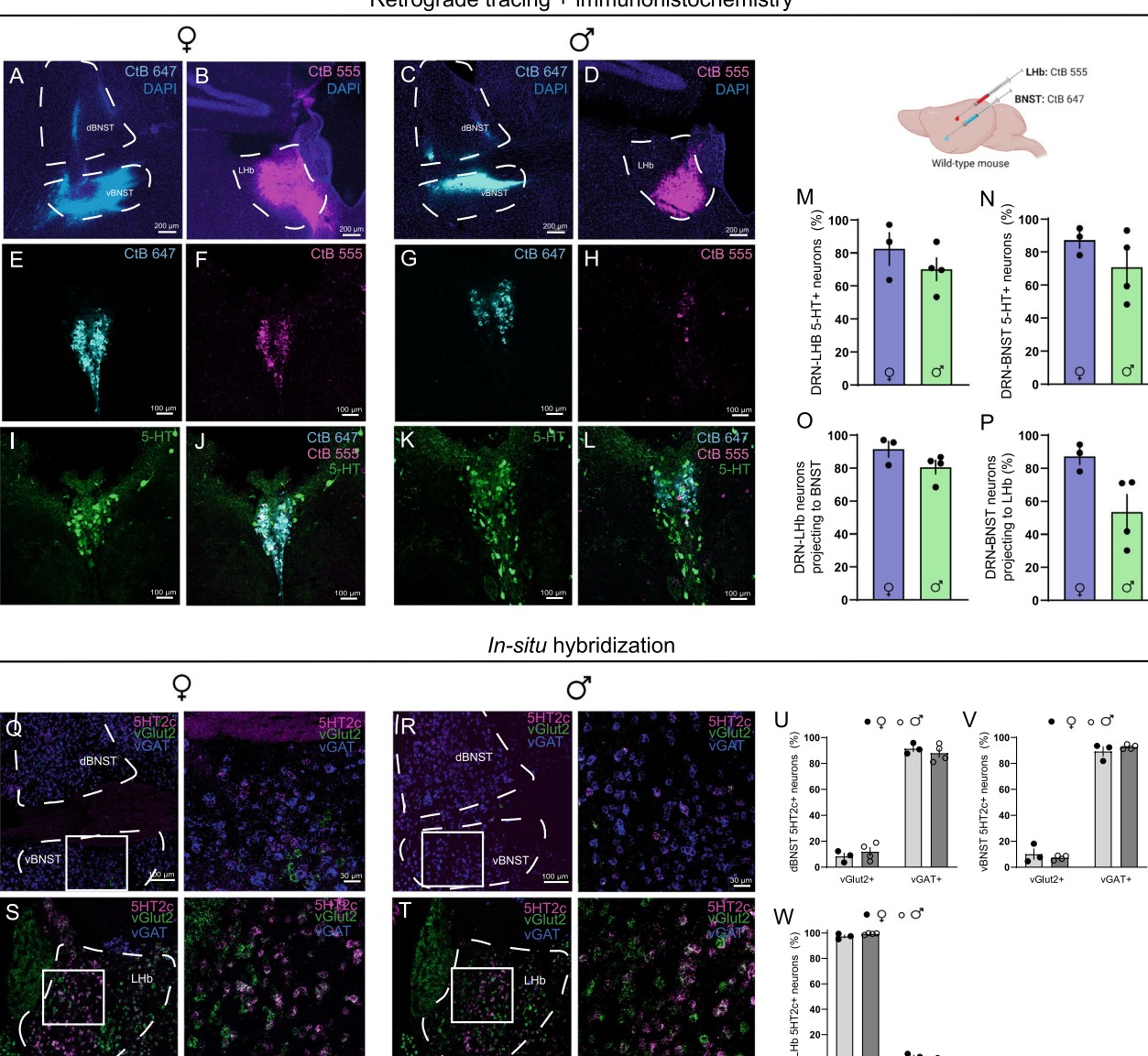

**Fig. 3 | Anatomical and neurochemical characterization of DRN-LHb and DRN-BNST circuits. A** Representative CtB 647 infection in BNST, females. Similar images were obtained in $n = 3$ mice. **B** Representative CtB 555 infection, females. Similar images were obtained in $n = 3$ mice. **C** Representative CtB 647 infection in BNST, males. Similar images were obtained in $n = 4$ mice. **D** Representative CtB 555 infection in LHb, males. Similar images were obtained in $n = 4$ mice. **E** Representative CtB 647 labeling in DRN, females. Similar images were obtained in $n = 3$ mice. **F** Representative CtB 555 labeling in DRN, females. Similar images were obtained in $n = 3$ mice. **G** Representative CtB 647 infection in BNST, males. Similar images were obtained in $n = 4$ mice. **H** Representative CtB 555 infection, males. Similar images were obtained in $n = 4$ mice. **I** Representative 5HT labeling in DRN, females. Similar images were obtained in $n = 3$ mice. **J** Representative labeling in DRN for CtB 647, CtB 555, and 5HT, females. Similar images were obtained in $n = 3$ mice. **K** Representative 5HT labeling in DRN, males. Similar images were obtained in $n = 4$ mice. **L** Representative labeling in DRN for CtB 647, CtB 555, and 5HT, males.

Similar images were obtained in $n = 4$ mice. **M** Percent of DRN-LHb neurons positive for 5HT **N** Percent of DRN-BNST neurons positive for 5HT. **O** Percent of CtB 555 neurons that co-express CtB 647 in DRN. **P** Percent of CtB 647 neurons that co-express CtB 555 in DRN. **Q** In-situ hybridization for 5HT$_{2c}$, vGAT, and vGlut2 in BNST, females. Similar images were obtained in $n = 3$ mice. **R** In-situ hybridization for 5HT$_{2c}$, vGAT, and vGlut2 in BNST, males. Similar images were obtained in $n = 4$ mice. **S,** In-situ hybridization for 5HT$_{2c}$, vGAT, and vGlut2 in LHb, females. Similar images were obtained in $n = 3$ mice. **T** In-situ hybridization for 5HT$_{2c}$, vGAT, and vGlut2 in LHb, males. Similar images were obtained in $n = 4$ mice. **U** Dorsal BNST 5HT2c neuron overlaps with vGAT or vGlut2. **V** Ventral BNST 5HT2c neuron overlaps with vGAT or vGlut2. **W** LHb 5HT2c neuron overlap with vGAT or vGlut2. $n = 3$ males, $n = 4$ females, 2 slices/mouse. No statistical comparisons were performed on this data as they are intended to be descriptive. All data are represented as mean ± SEM. Source data are provided as a Source Data file. Created with Biorender.com.

startle responses in male DiD mice persisted out to this four week time point. Remarkably, when we tested the same male mice again at eight weeks post-DiD exposure, we still observed an enhancement in their startle responses (Fig. S4). These data indicate that binge alcohol consumption uniquely alters affective behaviors in males and females, with male startle behavior being enhanced for a period far exceeding that of actual alcohol exposure.

To investigate whether abstinence from binge alcohol consumption is associated with altered expression of 5HT$_{2c}$, we next subjected a separate cohort of mice to the DiD paradigm and collected brain tissue punches of the LHb and the BNST for qPCR-based analysis of *Htr2c* mRNA expression (Fig. 5O–T). While DiD did not alter *Htr2c* mRNA levels in the BNST (Fig. 5T), it induced an increase in *Htr2c* mRNA levels in the LHb, which post-hoc analysis revealed was particularly

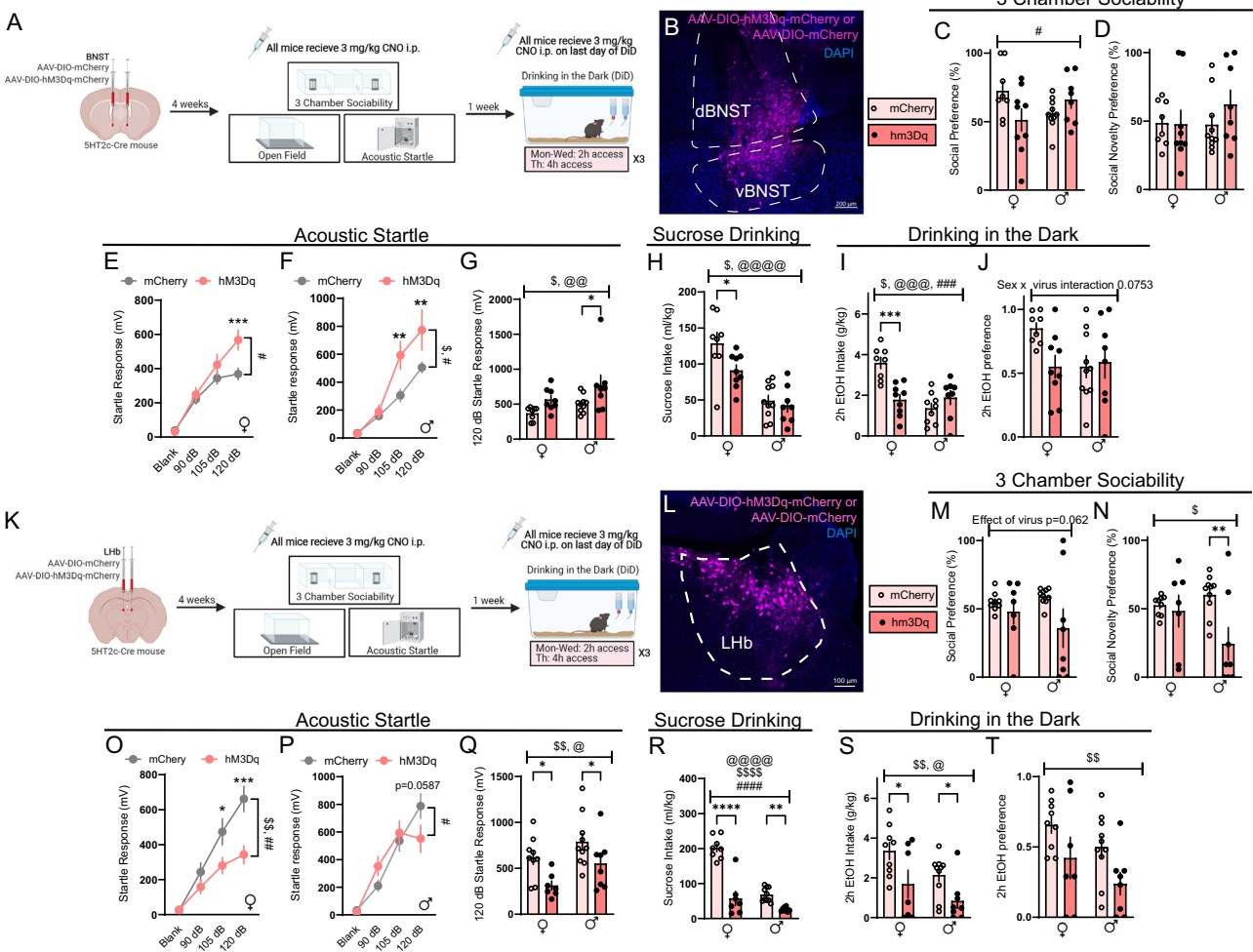

**Fig. 4 | Chemogenetic activation of LHb$_{5HT2c}$ or BNST$_{5HT2c}$ neurons modulates affective behaviors and binge alcohol consumption. A** Surgical schematic and experimental timeline for BNST$_{5HT2c}$ chemogenetic activation experiments. **B** Representative viral infection in BNST. Similar viral infection was reproduced in $n = 35$ mice. **C** Social preference in the 3-chamber sociability test, $p_{virusXsex} = 0.0192$. **D** Social novelty preference in the 3-chamber sociability test. **E** Acoustic startle behavior in females (10 trials/dB), two-way repeated-measures ANOVA followed by Holm-Sidak's post-hoc, $p_{virus} = 0.0814$, $p_{virusXdB} = 0.0105$. 3. **F** Acoustic startle behavior in males (10 trials/dB), two-way repeated-measures ANOVA followed by Holm-Sidak's post-hoc, $p_{virus} = 0.0246$, $p_{virusXdB} = 0.0111$. **G** Acoustic startle behavior 120 dB only (10 trials), $p_{virus} = 0.0058$, $p_{sex} = 0.037$. **H** Sucrose consumption, $p_{virus} = 0.0387$, $p_{sex} = 0.0001$. **I** 2 h Alcohol intake in DiD, $p_{virus} = 0.0303$, $p_{sex} = 0.0007$, $p_{sexXvirus} = 0.0003$. **J** 2 h Alcohol preference in DiD. **K** Surgical schematic and experimental timeline for LHb$_{5HT2c}$ chemogenetic activation experiments. **L** Representative viral infection in the LHb. Similar viral infection was reproduced in $n = 31$ mice. **M** Social preference in the 3-chamber sociability test. **N** Social novelty preference in the 3-chamber sociability test, $p_{virus} = 0.0186$.

**O** Acoustic startle behavior, females (10 trials/dB), two-way repeated measures ANOVA followed by Holm-Sidak's post-hoc, $p_{virus} = 0.0084$, $p_{virusXdB} = 0.0051$. **P** Acoustic startle behavior, males (10 trials/dB), two-way repeated measures ANOVA followed by Holm-Sidak's post-hoc, $p_{virusXdB} = 0.012$. **Q** Acoustic startle behavior, 120 dB only (10 trials), $p_{sex} = 0.0221$, $p_{virus} = 0.0031$. **R** Sucrose consumption, $p_{virus} < 0.0001$, $p_{sex} < 0.0001$, $p_{virus \times sex} < 0.0001$. **S** 2 h Alcohol consumption in DiD, $p_{virus} = 0.0026$, $p_{sex} = 0.0301$. T, 2 h Alcohol preference in DiD, $p_{virus} = 0.0099$, $p_{sex} = 0.0696$. BNST, Females: mcherry $n = 8$ mice, hM3Dq $n = 9$ mice; Males: mCherry $n = 10$ mice, hM3Dq $n = 8$ mice. LHb, Females: mCherry $n = 9$ mice, hM3Dq $n = 7$ mice; Males: mCherry n = 9 mice, hM3Dq n = 8 mice. Unless otherwise stated, statistical comparisons were performed using a two-way ANOVA followed by Holm-Sidak's post-hoc. All data are presented as mean ± SEM. \$ denotes effect of virus, @ denotes effect of sex, # denotes interaction (sex × virus or virus × stimulus strength), * denotes post-hoc $p < 0.05$, ** denotes post-hoc $p < 0.01$, *** denotes post-hoc $p < 0.001$, **** denotes post-hoc $p < 0.0001$. CNO: clozapine-n-oxide. Source data are provided as a Source Data file. Created with Biorender.com.

pronounced in males (Fig. 5S). Thus, affective disturbances induced by DiD are accompanied by increased expression of 5HT$_{2c}$ in the LHb, at least at the mRNA level.

**Binge alcohol consumption enhances the intrinsic excitability of LHb$_{5HT2c}$ neurons in both sexes**
Previous studies have demonstrated that alcohol exposure can enhance the activity of neurons in the LHb and the BNST during subsequent abstinence[24,25]. To investigate whether this is specifically the case for 5HT$_{2c}$-containing neurons in these regions, we subjected 5HT$_{2c}$-Ai9 reporter mice to DiD and performed whole-cell patch clamp electrophysiology in the LHb and the BNST of the same mice (Fig. 6A).

Of note, recordings in BNST$_{5HT2c}$ neurons were performed in the ventral BNST, while recordings in LHb$_{HT2c}$ neurons were performed in both the medial and lateral aspects of the LHb. We did not observe any differences in resting membrane potential (RMP) of LHb$_{5HT2c}$ neurons between water and DiD groups or between males and females (Fig. 6E). However, female BNST$_{5HT2c}$ neurons had on average higher RMPs than those of males (Fig. 6K). Rheobase (current required to elicit an action potential) (Fig. 6G, M), action potential threshold (Fig. 6H, N), or number of current-elicited action potentials of BNST$_{5HT2c}$ neurons (Fig. 6O, P) did not differ between groups. However, DiD induced a trend for a reduction in voltage-elicited current selectively in females that was particularly pronounced at higher voltages (Fig. 6S). For

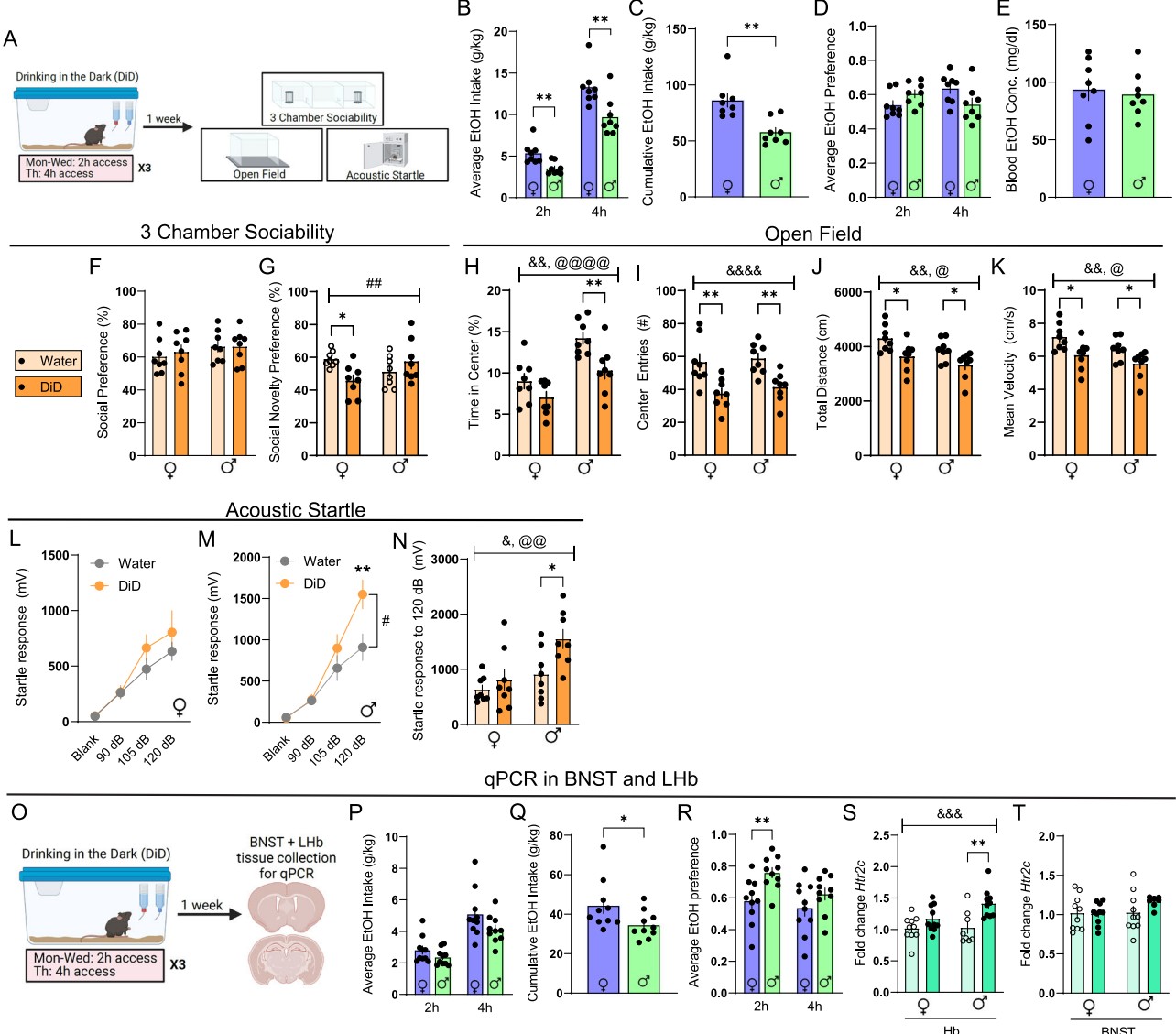

**Fig. 5 | DiD induces unique affective disturbances in males and females.**
**A** Experimental timeline for DiD behavioral studies. **B** Average alcohol intake in DiD, two-tailed *Student's* unpaired t-tests, $p_{2h} = 0.0042$, $p_{4h} = 0.0069$**C** Cumulative alcohol intake in DiD, two-tailed *Student's* unpaired t-test, $p = 0.0017$. **D** Average alcohol preference in DiD, two-tailed *Student's* unpaired t-tests, $p_{2h} = 0.0841$, $p_{4h} = 0.0685$. **E** Blood alcohol concentration immediately following 4h of DiD, two-tailed *Student's* unpaired t-test. **F** Social preference in the 3-chamber sociability test. **G** Social novelty preference in the 3-chamber sociability test, $p_{sexXDiD} = 0.008$. **H** Percent time in center of open field, $p_{DiD} = 0.0019$, $p_{sex} < 0.0001$. **I** Center entries open field, $p_{DiD} < 0.0001$. **J** Total distance in open field, $p_{DiD} = 0.0013$, $p_{sex} = 0.0366$. **K** Mean velocity in open field, $p_{DiD} = 0.0013$, $p_{sex} = 0.0366$. **L** Acoustic startle behavior, females (10 trials/dB), two-way repeated measures ANOVA. **M** Acoustic startle behavior, males (10 trials/dB), two-way repeated measures ANOVA followed by Holm-Sidak's post-hoc, $p_{DiD} = 0.0821$, $p_{DiDXdB} = 0.0055$. **N** Acoustic startle behavior, 120 dB only (10 trials), $p_{DiD} = 0.0164$, $p_{sex} = 0.0034$. **O** Experimental timeline for DiD qPCR studies. **P** Average alcohol intake in DiD, two-tailed Student's

unpaired t tests, $p_{2h} = 0.1589$, $p_{4h} = 0.1012$. **Q** Cumulative alcohol intake in DiD, two-tailed *Student's* unpaired t-test, $p = 0.0448$. **R** Average alcohol preference in DiD, two-tailed *Student's* unpaired t tests, $p_{2h} = 0.0054$, $p_{4h} = 0.1964$. **S** Relative expression of *Htr2c* mRNA in Hb, $p_{DiD} = 0.0005$, $p_{sex} = 0.0902$, $p_{DiDXsex} = 0.1012$. **T** Relative expression of *Htr2c* mRNA in BNST. For panels 3B-3N, Females: water $n = 8$ mice, DiD $n = 8$ mice; Males: water $n = 8$ mice, DiD $n = 8$ mice. For panels 3P–3R, Females: $n = 10$ mice; Males: $n = 10$ mice. For panels 3S and 3T, Females: water $n = 10$ mice/3 replicates, DiD $n = 10$ mice/3 replicates; Males: water $n = 10$ mice/3 replicates, DiD $n = 10$ mice/3 replicates. Unless otherwise stated, statistical comparisons were performed using two-way ANOVA followed by Holm-Sidak's post-hoc. All data are presented as mean ± SEM. & denotes effect of DiD (&$p < 0.05$, &&$p < 0.01$, &&&$p < 0.001$, &&&&$p < 0.0001$), @ denotes effect of sex (@$p < 0.05$, @@$p < 0.01$, @@@$p < 0.001$, @@@@$p < 0.0001$), # denotes interaction (sex × DiD or DiD × stimulus strength, #$p < 0.01$, ##$p < 0.01$), and * denotes post-hoc $p < 0.05$, ** denotes post-hoc $p < 0.01$. Source data are provided as a Source Data file. Created with Biorender.com.

LHb$_{5HT2c}$ neurons, males displayed an increased number of current-elicited action potentials in DiD mice compared to water mice (Fig. 6J). Although females did not display differences in rheobase between water and DiD groups when considering the entire LHb, when only medial LHb neurons were included in the analysis it was decreased in DiD mice, indicating greater excitability (Fig. 6U, V). Thus, females may display region-specific increases in LHb$_{5HT2c}$ neuronal excitability,

whereas males display these increases across LHb$_{5HT2c}$ neurons. This region-specificity of DiD-induced adaptations in females may explain why we did not observe a strong effect of DiD on total levels of LHb *Htr2c* mRNA in females, as our tissue punches included the entirety of the LHb. Notably, there was no effect of DiD the proportions of tonically active, silent, intermittently active/bursting, or depolarization-blocked cells (silent at >−40 mV) in either region (Fig. 6W, Z).

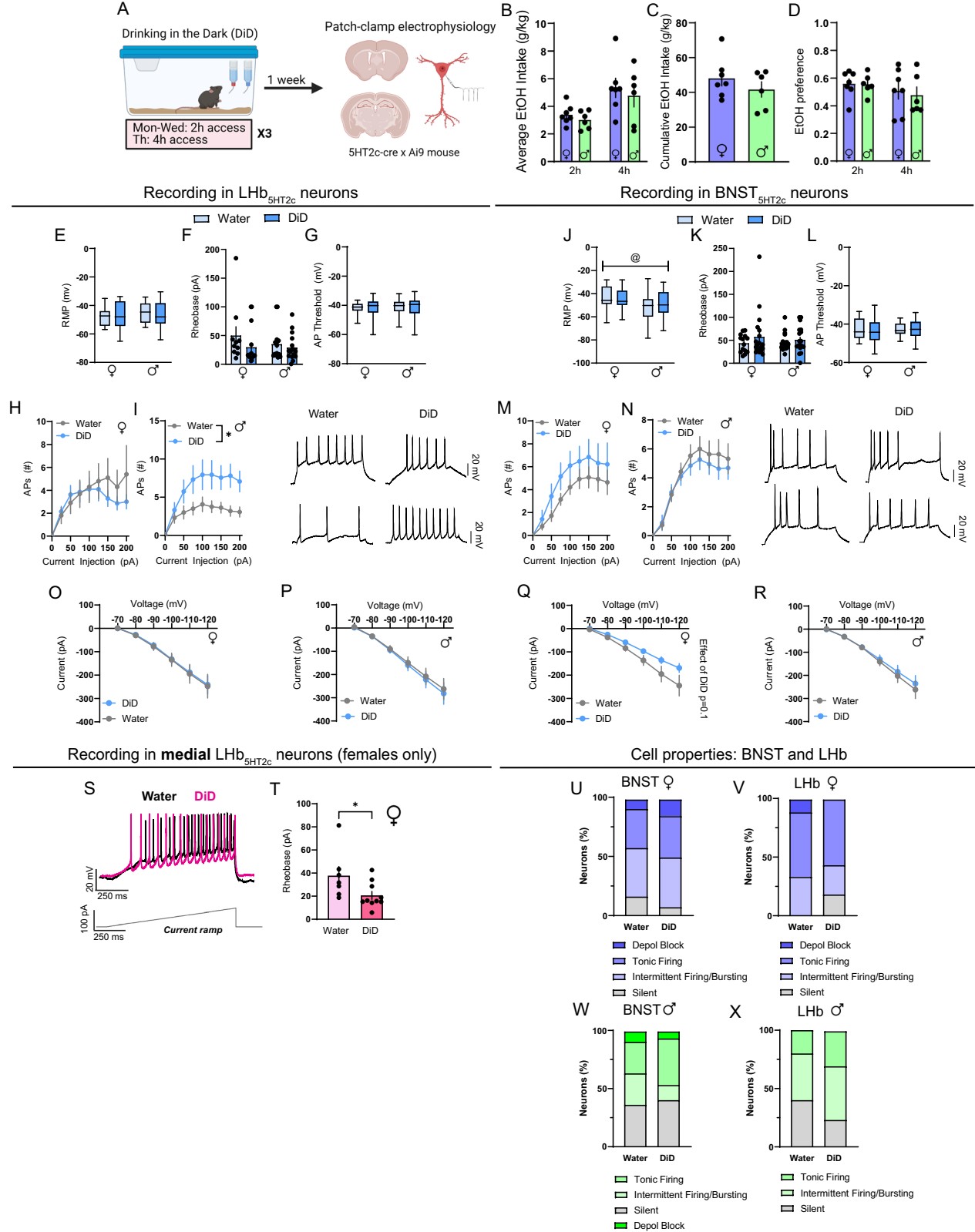

## Binge alcohol consumption modulates responses of DRN-LHb$_{5HT2c}$ and DRN-BNST$_{5HT2c}$ circuits to affective stimuli

In our next set of experiments, we sought to determine if social and arousal behaviors impacted by binge alcohol consumption are accompanied by alterations in the responses of LHb$_{5HT2c}$ and BNST$_{5HT2c}$ neurons to relevant stimuli. Following their first round of recordings in an alcohol-naïve state, the same mice used in the fiber

photometry experiments of Fig. 1 were randomly assigned to DiD or water groups. After three weeks of DiD or water exposure followed by one week of abstinence, we again performed either GCaMP7f or GRAB-5HT recordings in LHb$_{5HT2c}$ and BNST$_{5HT2c}$ during behavior (Figs. 7, S5–9, S13). DiD had no effect on female BNST$_{5HT2c}$ or LHb$_{5HT2c}$ calcium responses to acoustic startle stimuli (Fig. 7M, N, Q, R), but induced a trend towards increased LHb$_{5HT2c}$ activation upon

**Fig. 6 | DiD induces physiological adaptations in LHb_{5HT2c} and BNST_{5HT2c}.**
**A** Experimental timeline for DiD electrophysiology studies. **B** Average alcohol intake in DiD, two-way Student's unpaired t-tests. **C** Cumulative alcohol intake in DiD, two-way Student's unpaired $t$ test. **D** Average alcohol preference in DiD, two-way Student's unpaired $t$ tests. **E** Resting membrane potential (RMP) in LHb_{5HT2c} neurons. **F** LHb_{5HT2c} rheobase. **G** LHb_{5HT2c} action potential (AP) threshold.
**H** LHb_{5HT2c} elicited APs, females, two-way repeated measures ANOVA. **I** LHb_{5HT2c} elicited APs, males, two-way repeated measures ANOVA followed by Holm-Sidak's post-hoc, $p_{DiD} = 0.0236$, $p_{DiDXcurrent} = 0.0512$. **J** Average RMP in BNST_{5HT2c} neurons, $p_{sex} = 0.0190$. **K** BNST_{5HT2c} rheobase. **L** BNST_{5HT2c} AP threshold. **M** BNST_{5HT2c} elicited APs, females, two-way repeated measures ANOVA. **N** BNST_{5HT2c} elicited APs, males, two-way repeated measures ANOVA. **O** Voltage−Current plot for LHb_{5HT2c} females, two-way repeated measures ANOVA. **P** Voltage−Current plot for LHb_{5HT2c} males, two-way repeated measures ANOVA. **Q** Voltage−current plot for BNST_{5HT2c} females, two-way repeated measures ANOVA, $p_{DiD} = 0.1032$, $p_{DiDXvoltage} = 0.0701$.
**R** Voltage−Current plot for BNST_{5HT2c} males, two-way repeated measures ANOVA.

**S** Representative trace in medial LHb_{5HT2c} during rheobase test. **T** Quantification of rheobase values in medial LHb_{5HT2c} in females, two-way Student's unpaired $t$ test, $p = 0.0157$ (water $n = 5$ mice/7 cells; DiD $n = 5$ mice/10 cells). **U** Breakdown of cell properties at RMP, BNST females. **V** Breakdown of cell properties at RMP, LHb females. **W** Breakdown of cell properties at RMP, BNST males. **X** Breakdown of cell properties at RMP, LHb males. Depol block = silent at greater than −40 mV. BNST Females: water $n = 6$ mice/10 cells, DiD $n = 6$ mice/19 cells; BNST Males: water $n = 5$ mice/16 cells, DiD $n = 5$ mice/16 cells. LHb Females: water $n = 7$ mice/14 cells, DiD $n = 6$ mice/19 cells; LHb Males: water $n = 6$ mice/19 cells, DiD $n = 6$ mice/19 cells. Unless otherwise stated, statistical comparisons were performed using two-way ANOVA followed by Holm-Sidak's post-hoc. All data except for box plots are represented as mean ± SEM. For box plots, center line is the mean, box limits are 25th–75th percentiles, whiskers are min to max. & denotes effect of DiD, @ denotes effect of sex, # denotes interaction (sex × DiD or DiD × current), and * denotes post-hoc $p < 0.05$. Source data are provided as a Source Data file. Created with Biorender.com.

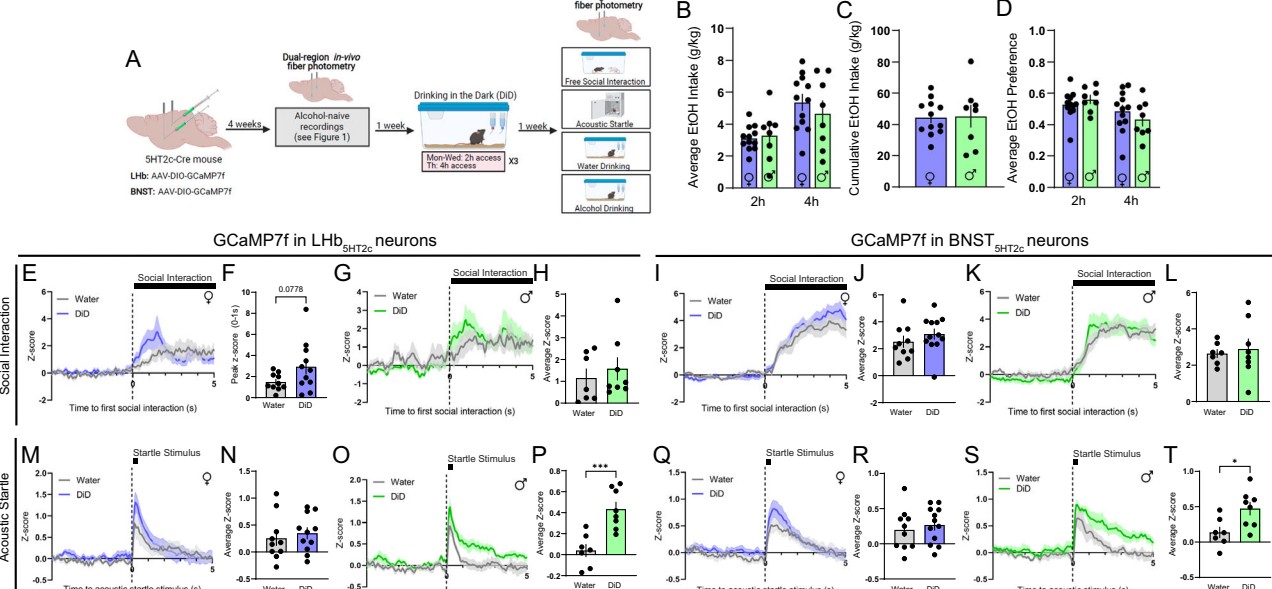

**Fig. 7 | DiD modulates the calcium responses of LHb_{5HT2c} and BNST_{5HT2c} neurons to affective stimuli.** **A** Surgical schematic and experimental timeline for LHb_{5HT2c} and BNST_{5HT2c} GCaMP fiber photometry experiments in DiD-exposed mice. **B** Average alcohol consumption in DiD ($n = 12$ females, $n = 8$ males). **C** Cumulative alcohol intake in DiD ($n = 12$ females, $n = 8$ males). **D** Average alcohol preference in DiD ($n = 12$ females, $n = 8$ males). **E** Peri-event plot of LHb_{5HT2c} GCaMP activity during the first interaction with a novel social target, females (water $n = 10$ mice/1 trial, DiD $n = 11$ mice/1 trial). **F** Peak z-score of LHb_{5HT2c} GCaMP signal for 0–1 s post interaction, females, Mann–Whitney $U$ test (water $n = 10$ mice/1 trial, DiD $n = 11$ mice/1 trial). **G** Peri-event plot of LHb_{5HT2c} GCaMP activity during the first interaction with a novel social target, males (water $n = 7$ mice/1 trial, DiD $n = 8$ mice/1 trial). **H** Average z-score of LHb_{5HT2c} GCaMP signal for 0–5 s post interaction, males (water $n = 7$ mice/1 trial, DiD $n = 8$ mice/1 trial). **I** Peri-event plot of BNST_{5HT2c} GCaMP activity during the first interaction with a novel social target, females (water $n = 10$ mice/1 trial, DiD $n = 12$ mice/1 trial). **J** Average z-score of BNST_{5HT2c} GCaMP signal for 0–5 s post interaction, females (water $n = 10$ mice/1 trial, DiD $n = 12$ mice/1 trial). **K** Peri-event plot of BNST_{5HT2c} GCaMP activity during the first interaction with a novel social target, males (water $n = 7$ mice/1 trial, DiD n = 8 mice/1 trial). **L** Average z-score of BNST_{5HT2c} GCaMP signal for 0–5 s post interaction, males (water $n = 7$

mice/1 trial, DiD $n = 8$ mice/1 trial). **M** Peri-event plot of LHb_{5HT2c} GCaMP activity during the acoustic startle test, females (water $n = 10$ mice/10 trials, DiD $n = 11$ mice/10 trials). **N** Average z-score of LHb_{5HT2c} GCaMP signal for 0–5 s post acoustic startle stimulus, females (water $n = 10$ mice/10 trials, DiD $n = 11$ mice/10 trials). **O** Peri-event plot of LHb_{5HT2c} GCaMP activity during the acoustic startle test, males (water $n = 7$ mice/10 trials, DiD $n = 8$ mice/10 trials). **P** Average z-score of LHb_{5HT2c} GCaMP signal for 0–5 s post acoustic startle stimulus, males, $p = 0.0008$ (water $n = 7$ mice/10 trials, DiD $n = 8$ mice/10 trials). **Q** Peri-event plot of BNST_{5HT2c} GCaMP activity during the acoustic startle test, females (water $n = 10$ mice/10 trials, DiD $n = 12$ mice/10 trials). **R** Average z-score of BNST_{5HT2c} GCaMP signal for 0–5 s post acoustic startle stimulus, females (water $n = 10$ mice/10 trials, DiD $n = 12$ mice/10 trials). **S** Peri-event plot of BNST_{5HT2c} GCaMP activity during the acoustic startle test, males (water $n = 7$ mice/10 trials, DiD $n = 8$ mice/10 trials). **T** Average z-score of BNST_{5HT2c} GCaMP signal for 0–5 s post acoustic startle stimulus, males, $p = 0.0157$ (water $n = 7$ mice/10 trials, DiD $n = 8$ mice/10 trials). Unless otherwise stated, statistical comparisons were performed using two-way Student's unpaired $t$ tests. All data represented as mean ± SEM. *$p < 0.05$, **$p < 0.01$, ***$p < 0.0001$. Source data are provided as a Source Data file. Created with Biorender.com.

interaction with a novel social target (Fig. 7E, F). In addition, DiD exposure was associated with the emergence of a negative LHb_{5HT2c} calcium response to alcohol in females, suggestive of a negative reinforcement signal in these neurons as a consequence of binge alcohol drinking experience (Fig. S5). In males, DiD robustly

increased BNST_{5HT2c} and LHb_{5HT2c} calcium responses to acoustic startle stimuli (Fig. 7O, P, S, T). While this potentiation of GCaMP responses to startle in males was primarily driven by early trials in LHb_{5HT2c}, it was primarily driven by late trials in BNST_{5HT2c} (Fig. S7). Interestingly, DiD had no impact on 5-HT release onto BNST_{5HT2c} or

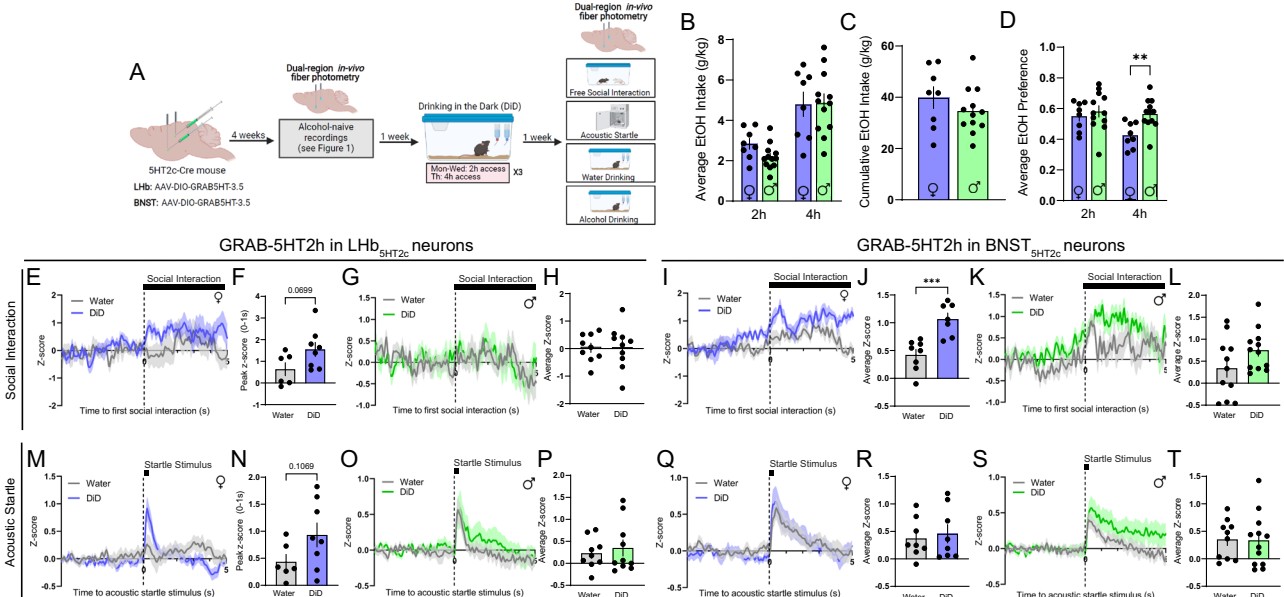

**Fig. 8 | DiD modulates serotonin release onto LHb_5HT2c and BNST_5HT2c neurons during exposure to affective stimuli. A** Surgical schematic and experimental timeline for LHb_5HT2c and BNST_5HT2c GRAB-5HT fiber photometry experiments in DiD-exposed mice. **B** Average alcohol consumption in DiD, $p_{2h} = 0.0532$ (Females $n = 8$ mice, Males $n = 12$ mice). **C** Cumulative alcohol intake in DiD, $p = 0.2897$ (Females $n = 8$ mice, Males $n = 12$ mice). **D** Average alcohol preference in DiD $p_{4h} = 0.0039$ (Females $n = 8$ mice, Males $n = 12$ mice). **E** Peri-event plot of LHb_5HT2c GRAB-5HT activity during the first interaction with a novel social target, females, (water $n = 6$ mice/1 trial, DiD $n = 8$ mice/1 trial). **F** Average z-score of LHb_5HT2c GRAB-5HT signal for 0–5 s post interaction, females, $p = 0.0699$ (water $n = 6$ mice/1 trial, DiD $n = 8$ mice/1 trial). **G** Peri-event plot of LHb_5HT2c GRAB-5HT activity during the first interaction with a novel social target, males (water $n = 9$ mice/1 trial, DiD $n = 10$ mice/1 trial). **H** Average z-score of LHb_5HT2c GRAB-5HT signal for 0–5 s post interaction, males (water $n = 9$ mice/1 trial, DiD $n = 10$ mice/1 trial). **I** Peri-event plot of BNST_5HT2c GRAB-5HT activity during the first interaction with a novel social target, females (water $n = 8$ mice/1 trial, DiD $n = 7$ mice/1 trial). **J** Average z-score of BNST_5HT2c GRAB-5HT signal for 0–5 s post interaction, females, $p = 0.0008$ (water $n = 8$ mice/1 trial, DiD $n = 7$ mice/1 trial). **K** Peri-event plot of BNST_5HT2c GRAB-5HT activity during the first interaction with a novel social target, males (water $n = 11$ mice/1 trial, DiD $n = 12$ mice/1 trial). **L** Average z-score of BNST_5HT2c GRAB-5HT signal for 0–5 s post interaction, males (water $n = 11$ mice/1 trial, DiD $n = 12$ mice/1 trial). **M** Peri-event plot of LHb_5HT2c GRAB-5HT activity during the acoustic startle test, females (water $n = 6$ mice/10 trials, DiD $n = 8$ mice/10 trials). **N** Peak z-score of LHb_5HT2c GRAB-5HT signal for 0–1 s post acoustic startle stimulus, females (water $n = 6$ mice/10 trials, DiD $n = 8$ mice/10 trials). **O** Peri-event plot of LHb_5HT2c GRAB-5HT activity during the acoustic startle test, males (water $n = 9$ mice/10 trials, DiD $n = 10$ mice/10 trials). **P** Average z-score of LHb_5HT2c GRAB-5HT signal for 0–5 s post acoustic startle stimulus, males (water $n = 9$ mice/10 trials, DiD $n = 10$ mice/10 trials). **Q** Peri-event plot of BNST_5HT2c GRAB-5HT activity during the acoustic startle test, females (water $n = 8$ mice/10 trials, DiD $n = 8$ mice/10 trials). **R** Average z-score of BNST_5HT2c GRAB-5HT signal for 0–5 s post acoustic startle stimulus, females (water $n = 8$ mice/10 trials, DiD $n = 8$ mice/10 trials). **S** Peri-event plot of BNST_5HT2c GRAB-5HT activity during the acoustic startle test, males (water $n = 11$ mice/10 trials, DiD $n = 12$ mice/10 trials). **T** Average z-score of BNST_5HT2c GRAB-5HT signal for 0–5 s post acoustic startle stimulus (water $n = 11$ mice/10 trials, DiD $n = 12$ mice/10 trials). Unless otherwise stated, statistical comparisons were performed using two-way Student's unpaired $t$ tests. All data represented as mean ± SEM. *$p < 0.05$, **$p < 0.01$, ***$p < 0.0001$. Source data are provided as a Source Data file. Created with Biorender.com.

LHb_5HT2c neurons in males for any of the behaviors tested (Figs. 8G, H, K, L, O, P, S, T, S6). Together with our electrophysiology findings suggesting increased intrinsic excitability of LHb_5HT2c neurons, these results indicate that the increased calcium responses to acoustic startle stimuli observed in DiD-exposed males are likely occurring post-synaptically and independently of acute fluctuations in 5-HT release. Females exposed to DiD, on the other hand, displayed a robust increase in 5-HT release onto BNST_5HT2c neurons in response to social interaction relative to water females (Fig. 8I–J), suggesting that binge alcohol consumption alters the pre-synaptic component of this circuit in the female BNST. Together with our electrophysiology findings in BNST_5HT2c, these results may suggest a model whereby alcohol consumption enhances 5-HT release onto BNST_5HT2c, which then results in a compensatory blunting of BNST_5HT2c excitability to influence subsequent behavior in females. We were not able to detect any significant alterations in 5-HT release onto LHb_5HT2c neurons as a consequence of DiD for either sex in any of the behaviors tested (Figs. 8, S8), but will note that DiD females displayed a trend towards increased LHb_5HT2c 5-HT release immediately following acoustic startle stimulus delivery and during social interaction (Fig. 8E, F, M, N). Taken together, these findings suggest that subcortical 5-HT release may be more subject to alcohol-induced modulation in female mice than in males.

## Genetic deletion of LHb or BNST 5HT_2c receptors partially normalizes DiD-induced social and arousal disturbances

To directly investigate the role of BNST and LHb 5HT_2c receptors in affective behaviors of alcohol-exposed vs. alcohol-naïve mice, we performed site-specific deletion of 5HT_2c receptors. We injected *Htr2c^lox/lox* mice in the BNST or the LHb with AAVs encoding cre fused to GFP or GFP alone. Expression of cre excises 5HT_2c receptor DNA from the genome of transduced cells via recombination at loxP sites[44], while expression of GFP alone keeps 5HT_2c expression intact (Figs. S10, S14). Four weeks following surgery, half of the mice were subjected to DiD and half continued to drink only water. One week following the last DiD session, mice were subsequently assessed in the 3-chamber sociability test, acoustic startle test, sucrose consumption test, and open field test (Figs. 9 and S11). Critically, in these experiments, enough time was given between surgery and the start of DiD for genetic recombination to occur, and thus 5HT_2c expression was repressed prior to alcohol exposure (Figs. 9, S10). With this in mind, we found that deletion of 5HT_2c in the BNST increased binge alcohol consumption in females, but had no effect in males (Fig. S11). This is consistent with what we observed in our chemogenetic experiments, where acute engagement of Gq signaling in BNST_5HT2c neurons reduced alcohol consumption in alcohol-experienced females but not males (Fig. 3). However, contrary to the findings of our chemogenetic experiments, deletion of 5HT_2c

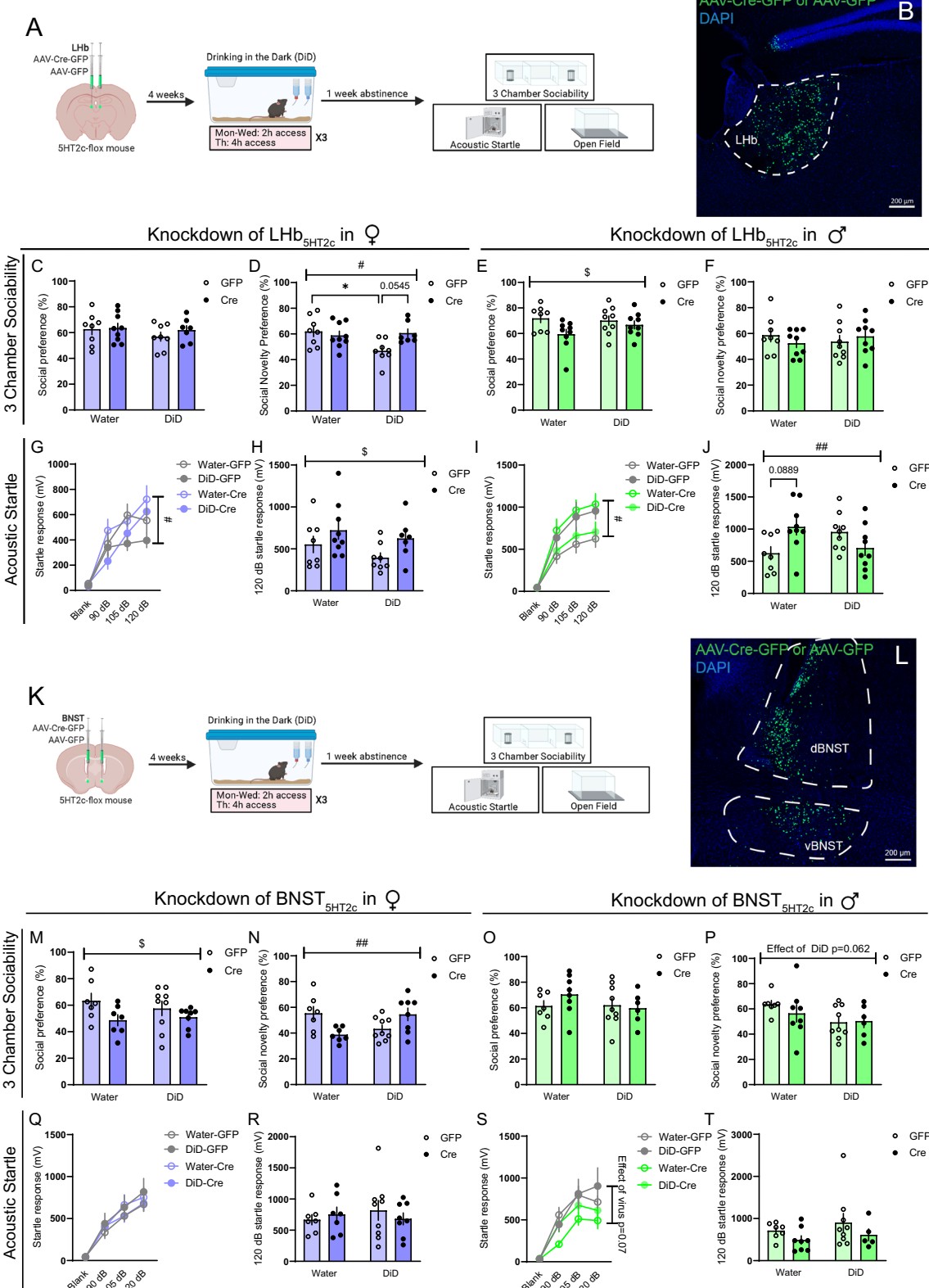

from the BNST had no impact on sucrose consumption in either sex (Fig. S11). This suggests that in females BNST $5HT_{2c}$ specifically regulates alcohol consumption and not simply the consumption of rewarding liquids per se. Deletion of $5HT_{2c}$ in the BNST had opposing effects on social recognition in alcohol-exposed versus alcohol-naïve females such that there was a significant interaction between virus treatment and DiD exposure. Specifically, alcohol-naïve cre females

displayed reduced social recognition compared to alcohol-naïve GFP females, but alcohol-exposed cre females displayed increased social recognition compared to alcohol-exposed GFP females (Fig. 9M, N). Interestingly, this manipulation also significantly reduced social preference in females regardless of alcohol history. There was no effect of $5HT_{2c}$ deletion in the BNST on social behavior in males (Fig. 9O, P) or on startle behavior in females (Fig. 9Q, R). However, regardless of

**Fig. 9 | Deletion of 5HT$_{2c}$ in the BNST or LHb partially normalizes alcohol-induced social and startle disturbances. A** Surgical schematic and experimental timeline for LHb 5HT$_{2c}$ deletion. **B** Representative viral infection for LHb 5HT$_{2c}$ deletion experiments. Similar viral infection was reproduced in $n = 66$ mice. **C** Social preference in the 3-chamber sociability test, females. **D** Social novelty preference in the 3-chamber sociability test, females, $p_{DiDXvirus} = 0.022$, $p_{DiD} = 0.0670$, $p_{virus} = 0.1352$. **E** Social preference in 3-chamber sociability test, males, $p_{virus} = 0.0334$. **F** Social novelty preference in 3-chamber sociability test, males. **G** Acoustic startle behavior, females (10 trials/dB), three-way repeated measures ANOVA followed Holm-Sidak's post-hoc, $p_{virusXdB} = 0.0112$, $p_{DiD} = 0.0908$, $p_{virusXdBXDiD} = 0.087$. **H** Acoustic startle behavior, 120 dB only, females (10 trials), $p_{virus} = 0.0417$. **I** Acoustic startle behavior, males (10 trials/dB), three-way repeated measures ANOVA followed by Holm-Sidak's post-hoc, $p_{virusXDiD} = 0.0104$, $p_{virusXdBXDiD} = 0.006$. **J** Acoustic startle behavior 120 dB only, males (10 trials), $p_{virusXDiD} = 0.0059$. **K** Surgical schematic and experimental timeline for BNST 5HT$_{2c}$ deletion. **L** Representative viral infection for BNST 5HT$_{2c}$ deletion experiments. Similar viral infection was reproduced in $n = 61$ mice. **M** Social preference in the 3-chamber sociability test, females, $p_{virus} = 0.0329$. **N** Social novelty preference in

the 3-chamber sociability test, females, $p_{DiDXvirus} = 0.0019$. **O** Social preference in 3-chamber sociability test, males. **P** Social novelty preference in 3-chamber sociability test, males, $p_{DiD} = 0.062$. **Q** Acoustic startle behavior, females (10 trials/dB), three-way repeated measures ANOVA, Holm-Sidak's post-hoc. **R** Acoustic startle behavior, 120 dB only, females (10 trials), three-way repeated measures ANOVA, Holm-Sidak's post-hoc. **S** Acoustic startle behavior, males (10 trials/dB), three-way repeated measures ANOVA followed by Holm-Sidak's post-hoc, $p_{virus} = 0.07$, $p_{virusXdB} = 0.1753$. **T** Acoustic startle behavior 120 dB only, males (10 trials). LHb females, GFP: water $n = 8$ mice, DiD $n = 8$ mice; Cre: water $n = 9$ mice, DiD $n = 6$ mice. LHb males, GFP: water $n = 8$ mice, DiD $n = 9$ mice; Cre: water $n = 9$ mice, DiD $n = 9$ mice. BNST females, GFP: water $n = 7$ mice, DiD $n = 9$ mice; DiD water $n = 7$ mice, DiD $n = 8$ mice. BNST males, GFP: water $n = 7$ mice, DiD $n = 9$ mice; Cre: water $n = 8$ mice, DiD $n = 6$ mice. Unless otherwise stated, statistical comparisons were performed using two-way ANOVA followed by Holm-Sidak's post-hoc. All data are presented as mean ± SEM. & denotes effect of DiD, # denotes interaction (DiD × virus), * denotes post-hoc effects. Source data are provided as a Source Data file. Created with Biorender.com.

---

alcohol experience, BNST 5HT$_{2c}$ deletion somewhat reduced startle behavior in males, although this effect was not statistically significant (Fig. 9S, T). It is notable that we did not observe the typical increase in startle behavior in male GFP mice following DiD exposure here, but this may be explained by the relatively low levels of baseline alcohol consumption in Htr$_{2c}$^{lox/lox} mice compared to wild-type C59BL6/J mice (see Fig. 4B–D). Exploratory behavior in the open field and locomotion were largely unaffected by BNST 5HT$_{2c}$ deletion, except for an interaction between virus treatment and DiD exposure on time spent in the center of the open field in females (Fig. S9). Together, these results suggest that 5HT$_{2c}$ receptor signaling in the BNST partially contributes to alcohol-induced affective disturbances in a sex-specific manner while also reducing binge alcohol consumption selectively in females.

Deletion of 5HT$_{2c}$ in the LHb had only weak effects on binge alcohol consumption in DiD, with both sexes showing trends towards increased overall alcohol preference in cre mice (Fig. S11) as well as interactions between viral treatment and day of DiD such that in later DiD sessions cre mice began to drink more than their GFP counterparts (Fig. S11). However, there was no effect of LHb 5HT$_{2c}$ deletion on sucrose consumption in either sex (Fig. S11). Similar to the effects in BNST, there was an interaction between virus treatment and DiD exposure on social recognition in females, suggesting that 5HT$_{2c}$ in the BNST and the LHb weakly promotes alcohol-induced disruptions in social recognition (Fig. 9C, D). Curiously, deletion of LHb 5HT$_{2c}$ reduced social preference in males regardless of alcohol history but had no effect on social recognition (Fig. 9E, F). In females, deletion of LHb 5HT$_{2c}$ increased startle behavior regardless of alcohol history, whereas in males there was an interaction between virus treatment and DiD exposure (Fig. 9G–J). Specifically, 5HT$_{2c}$ deletion in alcohol-naïve males increased startle behavior while in alcohol-exposed males it reduced startle behavior. Together, these results suggest that in both the BNST and the LHb, 5HT$_{2c}$ does not appear to be the predominant mechanism by which alcohol induces behavioral dysregulation, as receptor deletion only partially normalized social and startle behaviors. Nevertheless, 5HT$_{2c}$ in the BNST does appear to be a powerful regulator of alcohol intake in females.

### Chemogenetic inhibition of LHb$_{5HT2c}$ fully normalizes DiD-induced social and arousal disturbances

To determine whether acute silencing of LHb$_{5HT2c}$ or BNST$_{5HT2c}$ could also modulate DiD-induced social and arousal disturbances, we performed chemogenetic inhibition of LHb$_{5HT2c}$ or BNST$_{5HT2c}$. We injected male and female 5HT$_{2c}$-cre mice in the LHb or BNST with cre-dependent AAVs encoding either mCherry alone or the inhibitory DREADD hm4Di fused to mCherry (Fig. S15). Four weeks following surgery, we split mice into water or DiD groups and let half of them

consume alcohol freely for three cycles without engagement of hm4Di. On the last day of DiD, hm4Di and mCherry mice were treated with 3 mg/kg CNO prior to their drinking session. Interestingly, neither chemogenetic inhibition of BNST$_{5HT2c}$ nor LHb$_{5HT2c}$ impacted alcohol drinking in males. In females, however, chemogenetic inhibition of LHb$_{5HT2c}$ increased alcohol intake (Fig. S12). These effects in females were specific to alcohol, as chemogenetic inhibition of LHb$_{5HT2c}$ did not impact sucrose consumption (Fig. S12). Together with our 5-HT$_{2c}$ deletion studies, our findings indicate that in the BNST, 5-HT$_{2c}$ itself regulates alcohol consumption in females; in the LHb, however, the activity of 5-HT$_{2c}$-containing neurons regulates alcohol consumption in females in a manner that may be only partly dependent on 5-HT$_{2c}$ receptor signaling. Chemogenetic inhibition of BNST$_{5HT2c}$ during the 3-chamber sociability test did not modulate deficits induced by DiD in females (Fig. 10N), nor did it impact social behavior in males (Fig. 10O, P). However, social novelty preference was enhanced in LHb$_{5HT2c}$ hm4Di males regardless of alcohol history (Fig. 10F). Critically, inhibition of LHb$_{5HT2c}$ fully rescued DiD-induced social recognition deficits in female mice, indicating that increased activity of LHb$_{5HT2c}$ neurons drives DiD-induced social impairment. Inhibition of LHb$_{5HT2c}$ also fully normalized startle behavior in DiD males. Together, these results identify excessive activation of LHb$_{5HT2c}$ as a critical mechanism for promoting sex-specific behavioral disturbances induced by binge alcohol consumption. Unfortunately, our BNST$_{5HT2c}$ chemogenetic inhibition experiments were more difficult to interpret in the context of startle, as we observed a DiD-induced increase in startle in females, but not males. While chemogenetic inhibition of BNST$_{5HT2c}$ fully normalized startle behavior in DiD females, startle behavior was unaffected in males of either DiD or Water groups. Considered together with the results of our chemogenetic activation studies and our 5HT$_{2c}$ deletion studies, these results may simply reflect a floor effect of BNST$_{5HT2c}$ inhibition on startle behavior in cases when startle behavior is not enhanced by alcohol. The factors determining whether females develop enhanced startle behavior as a consequence of DiD should be investigated in future studies.

## Discussion

Previous anatomical and functional evidence suggests that the DRN is comprised of parallel sub-systems that differ in their input-output connectivity, physical localization within the DRN, neurotransmitter co-release properties, and roles in behavior[19,20,29,57–60]. Using a combination of anatomy and in-vivo biosensor measurement of 5-HT release, we found that many single 5-HT neurons in the caudal dorsal DRN target both the BNST and the LHb and appear to have coordinated 5-HT release in these regions. We demonstrate that this release is associated with increased calcium signaling in BNST$_{5H2Tc}$ and LHb$_{5HT2c}$

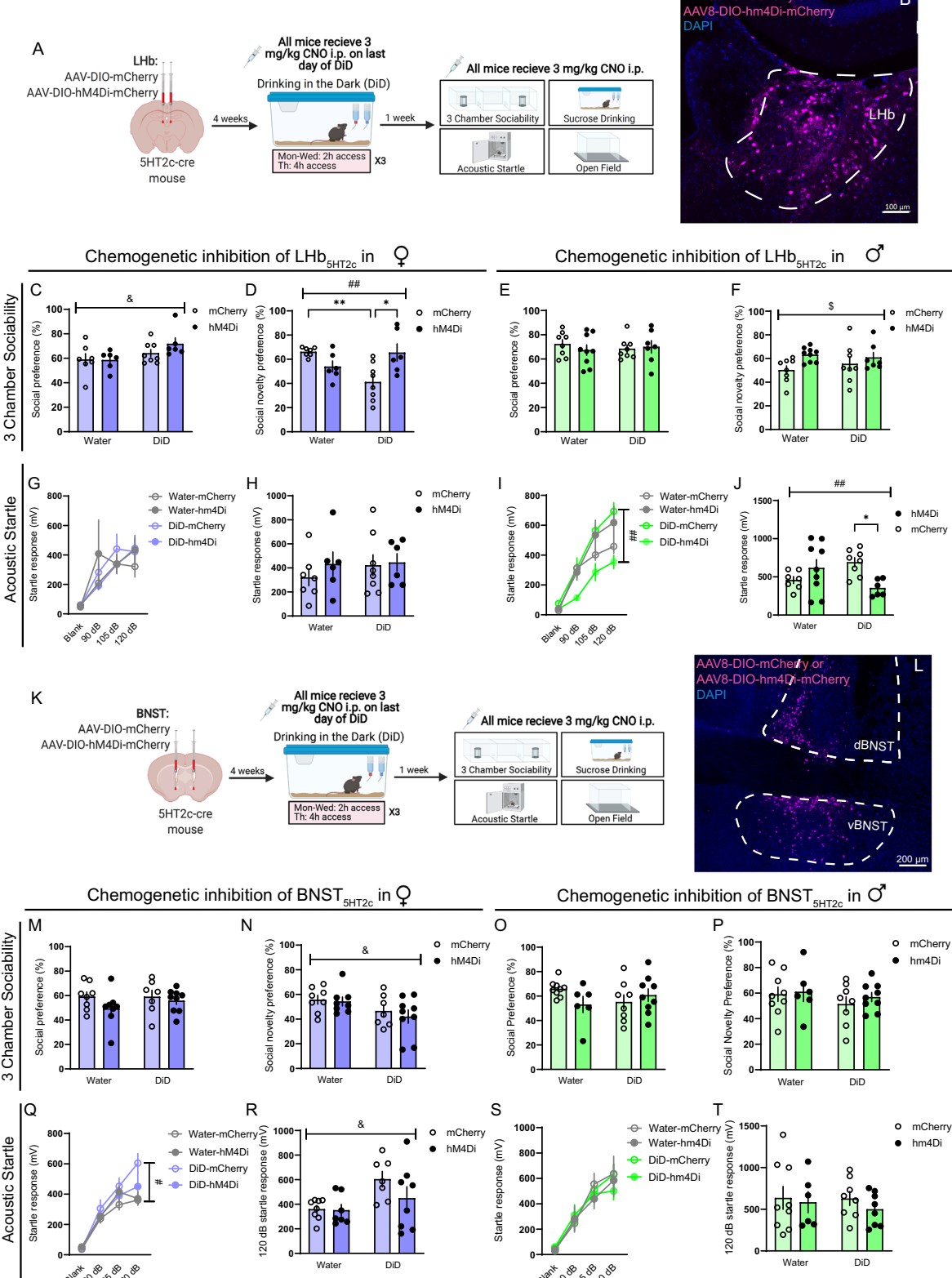

populations. Chemogenetic stimulation of BNST$_{5HT2c}$ and LHb$_{5HT2c}$ neurons in alcohol-naïve mice revealed many common functions of these neurons in promoting dysregulation of social and arousal behaviors. Perhaps reflecting previous data supporting the BNST as a sexually dimorphic structure[34], a number of sex differences were observed in the behavioral functions of BNST$_{5HT2c}$ neurons. We also found that binge alcohol consumption, which is associated with the

development of negative affective states during abstinence, alters DRN-BNST$_{5HT2c}$ and DRN-LHB$_{5HT2c}$ in-vivo circuit physiology distinctly in each sex, with males displaying enhanced calcium signaling in both BNST$_{5HT2c}$ and LHb$_{5HT2c}$ neurons in response to acoustic startle stimuli, and females displaying robust enhancement of 5-HT release onto BNST$_{5HT2c}$ neurons in response to novel social targets. Furthermore, DiD was associated with increased expression of LHb *Htr2c* mRNA and

**Fig. 10 | Chemogenetic inhibition of LHb$_{5HT2c}$ normalizes DiD-induced social and arousal disturbances. A** Surgical schematic and experimental timeline. **B** Representative image of viral infection in LHb. Similar viral infection was reproduced in $n = 59$ mice. **C** Social preference in the 3-chamber sociability test, $p_{DiD} = 0.0317$ **D** Social novelty preference in the 3-chamber sociability test, LHb females, $p_{DiD} = 0.2023$, $p_{virus} = 0.2472$, $p_{virusXDiD} = 0.0016$. **E** Social preference in 3-chamber sociability test, LHb males. **F** Social novelty preference in the 3-chamber sociability test, LHb males, $p_{virus} = 0.0298$. **G** Acoustic startle behavior, LHb females (10 trials/dB). **H** Acoustic startle behavior 120 dB only, LHb females (10 trials). **I** Acoustic startle behavior, LHb males (10 trials/dB), $p_{virus} = 0.1033$, $p_{virusXDiD} = 0.0028$, $p_{virusXdBXDiD} = 0.0043$. **J** Acoustic startle behavior 120 dB only, LHb males (10 trials), $p_{virusXDiD} = 0.0044$. **K** Surgical schematic and experimental timeline. **L** Representative image of viral infection in BNST. Similar viral infection was reproduced in $n = 64$ mice. **M** Social preference in 3-chamber sociability test, BNST females. **N** Social novelty preference in 3-chamber sociability test, BNST females, $p_{DiD} = 0.0253$. **O** Social preference in the 3-chamber sociability test, BNST males. **P** Social novelty preference in 3-chamber sociability test, BNST males. **Q** Acoustic startle behavior, BNST females (10 trials/dB), $p_{virus} = 0.1188$, $p_{virusXdB} = 0.0294$, $p_{DiDXdB} = 0.0459$, $p_{virusXdBXDiD} = 0.045$. **R** Acoustic startle behavior 120 dB only, BNST females (10 trials), $p_{DiD} = 0.0130$, $p_{virusXDiD} = 0.2662$. **S** Acoustic startle behavior, BNST males (10 trials/dB). **T** Acoustic startle behavior 120 dB only, BNST males, $p_{virus} = 0.1282$ (10 trials). LHb females, mCherry: water $n = 7$ mice, DiD $n = 8$ mice; hM4Di: water $n = 6$ mice, DiD $n = 6$ mice. LHb males, mCherry: water $n = 8$ mice, DiD $n = 8$ mice; hM4Di: water $n = 9$ mice, DiD $n = 7$ mice. BNST females, mCherry: water $n = 8$ mice, DiD $n = 7$ mice; hM4Di: water $n = 8$ mice, DiD $n = 9$ mice. BNST males, mCherry: water $n = 9$ mice, DID $n = 8$ mice; hM4Di water $n = 6$ mice, DiD $n = 9$ mice. All data are presented as mean ± SEM. & denotes effect of DiD, # denotes interaction (DiD x virus, #$p < 0.05$, ##$p < 0.01$), $ denotes effect of virus ($$p < 0.05$, $$$p < 0.01$), * denotes post-hoc $p < 0.05$, ** denotes $p < 0.01$. CNO: clozapine-n-oxide. Source data are provided as a Source Data file. Created with Biorender.com.

LHb$_{5HT2c}$ ex-vivo excitability, particularly in males. Genetic deletion of 5-HT$_{2c}$ in the LHb, and to a lesser extent the BNST, partially normalized social and startle disturbances induced by DiD. However, chemogenetic inhibition of LHb$_{5HT2c}$ neurons fully rescued sex-specific social and startle disturbances, indicating that 5-HT$_{2c}$ itself may only partially contribute to excessive activation of LHb$_{5HT2c}$ following alcohol exposure (see visual summary of findings in Fig. S16).

Our results suggest that LHb$_{5HT2c}$ neurons and BNST$_{5HT2c}$ neurons are activated by aversive stimuli (acoustic startle stimuli) and inhibited by rewarding stimuli (water or alcohol consumption) in both males and females. One exception to this appears to be interaction with a novel juvenile social target, which is purported to be a rewarding stimulus for both adult males and adult females[61,62]. Indeed, first contact with a juvenile social target in a neutral context (not the experimental mouse's home cage) elicited activation in both LHb$_{5HT2c}$ and BNST$_{5HT2c}$ neurons, although this activation was greater in the BNST for males compared to females. Recent non-cell-type specific fiber photometry studies performed in male mice indicate that LHb neurons are not notably modulated by non-aggressive interactions with familiar adult conspecifics[63], but in-vivo recordings of LHb neurons during interactions with juveniles have not been performed, nor have recordings in females.

To our knowledge, in-vivo recordings in the BNST have not previously been performed during non-aggressive social interactions in either sex. Crucially, for both sexes the degree of BNST$_{5HT2c}$ neuronal activation in response to the first interaction with a novel juvenile social target was greater than that of any other stimulus tested, suggesting that this may be reflective of a salience-driven signal, as opposed to a valence-driven signal. This is consistent with a recent report illustrating that a large proportion of individual DRN-BNST neurons respond to a variety of valenced stimuli, including social stimuli, electric shock, and contextual threat[64]. Only in our voluntary alcohol consumption fiber photometry experiments did we observe a notable divergence between patterns of 5-HT release onto LHb$_{5HT2c}$ and BNST$_{5HT2c}$ neurons and patterns of calcium activity in these neurons. In naïve females, alcohol consumption did not coincide with decreased calcium activity, yet 5-HT release was reduced during alcohol consumption (Figs. 1, 2). In males, however, both calcium activity and 5-HT release were reduced upon alcohol consumption. While the precise reasons for this are not clear, these results could reflect differences in post-synaptic 5-HT receptor signaling in females compared to males. More generally, these findings are in keeping with other studies supporting the BNST as a critical site for integration of salience and value, and extend these prior findings by demonstrating the in-vivo dynamics of 5-HT release and calcium activity during behavior.

Our chemogenetic activation studies in BNST$_{5HT2c}$ neurons highlight important sex differences in the function of Gq signaling on affective behaviors and binge alcohol consumption. Chemogenetic activation of BNST$_{5HT2c}$ neurons increased acoustic startle responses in both sexes, an effect that is in agreement with previous findings from our group showing that intra-BNST infusion of the 5HT$_{2c}$ agonist mCPP increases acoustic startle responses in male mice[65]. However, no other behavior tested was altered by chemogenetic activation of BNST$_{5HT2c}$ neurons in males. Binge alcohol consumption, sucrose consumption, and to a lesser degree social preference, were reduced by chemogenetic activation of BNST$_{5HT2c}$ neurons in females, indicating that these neurons play a more significant role in reward consumption in females compared to males. The potential mechanisms underlying these sex differences in reward consumption are numerous, but could be due to variations in the density or synaptic strength of downstream projections of BNST$_{5HT2c}$ neurons, perhaps to regions like the ventral tegmental area or the DRN.

In contrast to the effects of chemogenetic activation of BNST$_{5HT2c}$, chemogenetic activation of LHb$_{5HT2c}$ did not produce sex-specific effects on affective behaviors. Instead, in both sexes, this manipulation had an inhibitory effect on liquid reward consumption (sucrose and alcohol), responses to acoustic startle stimuli, social behavior, and exploratory behavior. There was no inhibition of locomotion induced by chemogenetic activation of LHb$_{5HT2c}$ neurons, indicating that these effects on affective behavior are not due to general behavioral inhibition. In regards to social behavior, these results are consistent with previous studies in males and extend the findings to females. Indeed, non-conditional chemogenetic activation of LHb neurons in male rats reduced social preference in the 3-chamber test in a recent study[66]. While no previous studies have investigated the effects of non-conditional chemogenetic activation of LHb neurons on alcohol-related behaviors, non-conditional chemogenetic inhibition of LHb neurons in male rats reduces withdrawal-related anxiety-like behavior[67] and voluntary alcohol consumption in an intermittent access model[68]. However, in a recent study employing chronic intermittent ethanol vapor followed by intermittent two-bottle choice access, non-conditional chemogenetic inhibition of LHb neurons in male rats did not impact alcohol drinking behaviors[69]. Our results are broadly concordant with these previous findings and suggest that LHb plays more of a role in behavior disruption induced by alcohol than alcohol consumption itself in males.

This study reveals an important distinction between the behavioral roles of LHb$_{5HT2c}$ neurons versus BNST$_{5HT2c}$ neurons: LHb$_{5HT2c}$ neurons serve to reduce active responses to stressors (reduced acoustic startle responses with chemogenetic activation), whereas BNST$_{5HT2c}$ neurons serve to increase active responses to stressors (increased acoustic startle responses with chemogenetic activation). Despite these opposing behavioral functions, our fiber photometry experiments showed that the delivery of acoustic startle stimuli increased calcium activity and 5-HT release in BNST$_{5HT2c}$ and LHb$_{5HT2c}$

neurons. Thus, LHb$_{5HT2c}$ and BNST$_{5HT2c}$ neurons drive opposing behavioral responses to stressful stimuli, yet these neurons display similar cellular responses to these stimuli. In human patients, reduced responses to acoustic startle stimuli are observed in patients with major depression, while increased responses to these stimuli are observed in patients with anxiety disorders[70]. Given that the LHb has been highly implicated in human major depression[71] and the BNST has been implicated in multiple sub-types of human anxiety disorders[30], it is then reasonable to speculate that regionally selective alterations of 5HT$_{2c}$-containing neurons in LHb and BNST could play a role these behavioral observations.

Previous studies suggest that neuronal activity is enhanced in both the BNST and the LHb as a consequence of alcohol exposure in males, in at least a partly 5HT$_{2c}$-dependent manner[24,25]. However, no studies to date have investigated the effects of alcohol consumption on 5HT$_{2c}$-mediated signaling in the LHb or the BNST in females. Withdrawal from intermittent access to alcohol in males increases cFos (a marker of neuronal activation) and 5HT$_{2c}$ protein expression in the LHb, and this increase in cFos is reversed with intra-LHb infusion of a 5HT$_{2c}$ antagonist[24]. Consistent with these findings, we found that abstinence from binge alcohol drinking increases both the expression of LHb *Htr2c* mRNA as well as neuronal excitability. However, both of these effects were more pronounced in males compared to females. In the male BNST, withdrawal from alcohol vapor exposure also increases cFos activation that can be reversed with systemic treatment with a 5HT$_{2c}$ antagonist[25]. Furthermore, although not performed selectively in BNST$_{5TH2c}$ neurons, slice electrophysiology results from this same study indicated that alcohol vapor exposure enhances the intrinsic excitability of BNST neurons in a 5HT$_{2c}$-dependent fashion in males. However, our slice electrophysiology recordings yielded no differences in BNST$_{5HT2c}$ neuronal excitability between water and DiD males. The discrepancy with previous studies in males could be the result of differences in degrees of intoxication achieved through voluntary DiD versus involuntary alcohol vapor exposure. In addition to displaying blunted voltage-current relationships compared to water females, DiD females had greater BNST$_{5HT2c}$ RMPs than those of males. This is somewhat concordant with findings in BNST neurons expressing corticotrophin-releasing factor, a subset of which express 5HT$_{2c}$[38]. In these neurons, binge alcohol consumption significantly increases the proportion of male cells that are in a depolarization block such that they begin to resemble the high proportions observed in females[72].

Our electrophysiology combined with our fiber photometry findings suggests that female mice may undergo only subtle changes in the physiological properties of BNST$_{5HT2c}$ neurons as a consequence of binge alcohol consumption. However, binge alcohol consumption appears to strongly increase 5-HT release onto BNST$_{5HT2c}$ neurons in females. This effect is stimulus-dependent, as the difference from water mice was only observed upon social interaction. Importantly, differences in stimulus-dependent activity between DiD and water mice were only observed for those behaviors which were sex-specifically disrupted as a consequence of DiD (see Fig. 4). Males, which displayed increased startle behavior as a result of DiD, showed heightened calcium responses to startle but not to social interaction. Females, which displayed reduced social behavior as a result of DiD, showed enhanced in-vivo 5-HT responses to social interaction but not to startle. Taken together, these results suggest that female subjects could be more responsive to manipulations of 5-HT release to improve alcohol-induced behavioral deficits and highlight the importance of examining both male and female subjects in preclinical research.

Perhaps the most striking findings from our 5HT$_{2c}$ deletion and chemogenetic inhibition experiments were those indicating that these manipulations could have opposing effects on social and arousal behaviors in alcohol-exposed vs. alcohol naïve individuals. In females, deletion of 5HT$_{2c}$ in the BNST or chemogenetic inhibition of BNST$_{5HT2c}$

partly normalized social recognition in DiD mice but disrupted it in water mice. In males, deletion of 5HT$_{2c}$ in the LHb or chemogenetic inhibition of LHb$_{5HT2c}$ normalized startle behavior in DiD mice but enhanced it in water mice. These data may suggest the existence of an inverted-U type relationship between levels of 5HT$_{2c}$ expression/ neural activity and specific affective behaviors in males and females such that too little 5HT$_{2c}$/neural activation in the case of water cre-treated (5HT$_{2c}$ deleted) mice or excessive 5HT$_{2c}$/neural activation in the case of DiD GFP-treated (5HT$_{2c}$ intact) mice dysregulate social and arousal processing.

Together with our chemogenetic experiments, our 5-HT$_{2c}$ deletion experiments also highlight important dissociations between cellular manipulations of Gq/Gi signaling broadly and manipulations directly affecting 5-HT$_{2c}$. For example, stimulation of Gq signaling in BNST$_{5HT2c}$ neurons increased startle behavior in females and stimulation of Gi signaling reduced it, but deletion of BNST 5-HT$_{2c}$ had no impact on startle behavior in females. Similarly, stimulation of Gq signaling in BNST$_{5HT2c}$ neurons in females reduced sucrose consumption, but deletion of 5-HT$_{2c}$ or stimulation of Gi signaling in BNST$_{5HT2c}$ had no impact on sucrose consumption. Furthermore, stimulation of Gq or Gi signaling in LHb$_{5HT2c}$ neurons had robust effects on alcohol consumption in females, but deletion of 5-HT$_{2c}$ did not markedly affect this behavior. These data may suggest that in the LHb and BNST, 5-HT$_{2c}$ itself plays less of an overall role in regulating social and arousal behaviors than in regulating the general activity of 5-HT$_{2c}$-containing neurons.

There are several limitations of the present work that warrant discussion and/or further investigation. Previous work from our laboratory suggests that heavy alcohol exposure is associated with increased excitability of ventral, but not dorsal, BNST (vBNST, dBNST) neurons in male mice[25]. Importantly, dorsal and ventral sub-regions of the BNST are known to be distinct in their circuit architecture and connectivity with regions outside of the BNST, though they are highly interconnected[34]. Therefore, we chose to perform our electrophysiology experiments specifically in the vBNST. While we also targeted our viral injections and optic fiber placements to the vBNST in subsequent photometry, receptor deletion, and chemogenetic experiments, we were not able to fully restrict viral expression to the vBNST and did observe spread to the dBNST. Thus, the vast majority of animals had both vBNST and dBNST viral expression (see Figs. S12–S14). Future studies should aim to determine whether there is a functional dissociation between dBNST and vBNST 5HT2c-containing neurons, perhaps by selecting non-overlapping outputs of dBNST and vBNST sub-regions and injecting a retrograde virus to achieve sub-region-selective viral expression. Other important limitations of this study involve the design of our fiber photometry experiments. In order to facilitate recordings during voluntary alcohol consumption under tethered conditions (attached to a patch cord), we were forced to briefly water-deprive experimental mice prior to testing. In an attempt to tease apart the rewarding effects of thirst quenching to the rewarding effects of alcohol, we also performed voluntary water drinking experiments under water deprivation conditions. However, it is possible that given increased thirst, the rewarding value of both water and alcohol were increased under these conditions. Whether the signals we observed would look different without prior water deprivation should be investigated in future studies, perhaps by performing more long-term home cage photometry recordings. Another potential caveat of our photometry experiments is that all mice were run through the battery of social, startle, and drinking behaviors twice, which could have impacted the results we observed in the second phase of testing. However, if this were the case, we would have expected to see changes in both DiD and Water groups. Rather, the photometry signals in Water mice during the second phase of testing closely resembled those obtained during the first phase, while DiD mice displayed marked differences from pre-DiD signals. The relatively

long time period (1 month) between testing phases also reduces the likelihood that the post-DiD signals were impacted by pre-DiD testing.

In conclusion, our study suggests that the LHb and BNST represent two critical targets of an aversive DRN 5-HT sub-system that are physiologically impacted by binge alcohol consumption in sex-specific ways. Functionally, it appears that the primary mechanism by which alcohol promotes sex-specific expression of negative affect is through increased activation of $LHb_{5HT2c}$, which is likely only partially $5HT_{2c}$-dependent. These data may have important implications for the development of novel, sex-specific treatments for AUD and comorbid mood disorders.

## Methods

### Animals
Male and female wild-type C57BL6/J (Stock #: 000664, Jackson Laboratories), transgenic $5HT_{2c}$-Cre (provided by Dr. Laura Heisler's lab, from Burke et al. 2016), transgenic $5HT_{2c} \times Ai9$ (Ai9 Stock #007909, $5HT_{2c}$-cre x Ai9 breeding performed in our animal facility), or transgenic $Htr_{2c}^{lox/lox}$ (provided by Dr. Joel Elmquist, University of Texas Southwestern) adult mice aged 2-5 months at the start of the experiment were used as experimental animals. Notably, all transgenic mice were of C57BL6/J background. Male and female albino C57BL6/J (Stock #: 000058, Jackson Laboratories) adolescent mice aged 5-6 weeks were used as social targets. $5HT_{2c}$-Cre, and $Htr2c^{lox/lox}$, and $5HT_{2c}$-Cre x Ai9 strains were bred in-house at UNC School of Medicine, while wild-type C57BL6/J and albino C5BL6/J mice were ordered from Jackson Laboratories and allowed to acclimate to our animal facility for one week prior to testing. Social target mice were group housed. Experimental mice were group housed for all experiments until the beginning of DiD, at which point they were single housed until experiment completion. All mice were housed in polycarbonate cages (GM500, Techniplast) under a 12:12 h reverse dark-light cycle where lights turned off at 7:00 A.M. Housing rooms were temperature (70–75 °F) and humidity (40–60%) controlled. Mice had ad-libitum access to food (Prolab Isopro RMH 3000, LabDiet) and water unless otherwise stated. All experiments were approved by the UNC School of Medicine Institutional Animal Care and Use Committee (IACUC) and in accordance with the NIH guidelines for the care and use of laboratory animals.

### Drinking in the Dark (DiD)
Experimental mice were singly housed at least three days before initiation of the DiD procedure and remained singly housed throughout the completion of the experiment. During this procedure, mice were given free access to both water and 20% (w/v) ethanol bottles in the home cage from 10:00 A.M. to 12:00 P.M. on Mondays, Tuesdays, and Wednesdays, and from 10:00 A.M. to 2:00 P.M. on Thursdays. At all other times, mice were given access to water alone. Water and ethanol bottles were weighed at the 2 h time point on Mondays, Tuesdays, and Wednesdays, and at the 2 h and 4 h time point on Thursdays. Ethanol and water bottle positions were alternated daily to account for any inherent side preference in the animals. This weekly DiD access schedule was repeated for 3 weeks total, after which mice remained abstinent until behavioral testing. Confirmation of binge levels of intoxication was performed by measuring the blood ethanol concentration of tail blood collected at the end of a 4 h DiD drinking session using the AM1 Analox Analyzer (Analox Instruments).

### 3-Chamber sociability test
The 3-chamber sociability test was performed using an apparatus consisting of a plexiglass rectangle measuring 20 cm × 40.5 cm × 22 cm that was divided into three equally-sized spaces by two panels with small square openings at the base to allow for movement between the chambers. Before the test began, social target mice were habituated for 10 min. to a 10.5 cm diameter metal holding cage consisting of vertical bars with gaps between them. Importantly, mice outside of the

holding cages can see, smell, and touch mice inside the holding cages but cannot freely interact wth them. Holding cages were placed in opposite corners of the two outermost chambers during all phases of the test (one in each of the two chambers). The test consisted of three 10 min. phases run one right after the other to total 30 min: 1. Habituation, 2. Social preference, and 3. Social novelty preference. For all phases, an experimental mouse was placed in the center chamber and allowed to move freely between chambers. In the habituation phase, the holding cages in the outermost chambers were empty. In the social preference phase, one holding cage contained a novel, same-sex, adolescent social target and the other holding cage contained a novel object (plastic toy mouse). In the social novelty preference phase, one holding cage contained the novel social target from the social preference phase (now familiar, on opposite side from social preference phase position) and the other holding cage contained a second novel social target. For all phases, the experimental mouse's movements were tracked and the time spent interacting with the holding cages was measured using Ethovision XT 14.0 (Noldus). The social preference ratio was determined by calculating: (time spent with novel mouse)/ (time spent with novel mouse + time spent with novel object) × 100 (for a % preference). The social novelty preference ratio was determined by calculating: (time spent with novel mouse)/(time spent with novel mouse + time spent with familiar mouse) × 100. Mice that did not enter both outermost chambers at least once were excluded from analysis for that phase of the test.

### Open field test
The open field test was performed in a plexiglass square arena with dimensions 50 cm × 50 cm × 40 cm. An experimental mouse was placed in the corner of the arena and allowed to freely explore for 10 min. Behavior was recorded with an overhead video camera. Mouse position data was acquired from videos using Ethovision XT (Noldus, Inc.). The total distance traveled, mean velocity, time spent in the 10 × 10 cm center zone, and number of entries to the center zone were calculated.

### Acoustic startle test
The acoustic startle test was performed with the SR-LAB Startle Response System (SD Instruments). The system consisted of a sound-attenuating isolation cabinet with dimensions 38 cm × 36 cm × 46 cm containing a small plastic cylindrical enclosure with dimensions 4 cm (diameter) × 13 cm (length). The isolation cabinet was lit by low-intensity white light throughout the test. At the start of the test, an experimental mouse was placed in the plastic enclosure and allowed to habituate for 5 min. After habituation, the mouse was presented with one of four different acoustic stimuli ten different times (for a total of 40 trials): 90 dB, 105 dB, 120 dB, and 0 dB. Acoustic stimuli were delivered for 40 ms each trial and the startle response was measured for 200 ms following stimulus delivery. Startle stimuli were delivered in a random order with random inter-stimulus intervals lasting 30–50 s. For each mouse, maximum responses for each stimulus type were averaged across the ten trials.

### Home cage sucrose consumption test
The sucrose consumption test was performed in the home cage of singly-housed experimental animals. Mice were given 4 hs of access to both water and a 2% (w/v) sucrose solution and bottles were weighed at the 2 h and 4 h time points. Sucrose consumption was normalized to body weight (ml/kg consumed).

### Stereotaxic Surgery
Adult mice (>7 weeks of age) were anesthetized with isoflurane (1–3%) in oxygen (1–2 L/min) and positioned in a stereotaxic frame using ear cup bars (Kopf Instruments). The scalp was sterilized with 70% ethanol and betadine and a vertical incision was made before using a drill to

burr small holes in the skull directly above the injection targets. Using a 1 μl Neuros Hamilton Syringe (Hamilton, Inc.), viruses were then microinjected at a 0° angle into the LHb (mm relative to bregma: AP: −1.5, ML: ± 0.5, DV: −2.95) and/or the BNST (mm relative to bregma: AP: + 0.7, ML: ± 0.9, DV: −4.60) at a volume of 200 nl of virus per injection site. For fiber photometry experiments, optic fibers were implanted in the LHb and BNST at the same DV as the viral injections. Optic fibers were secured to the skull using Metabond dental cement (Parkell, Inc.). For chemogenetics experiments, $5HT_{2c}$-Cre mice were injected with either AAV8-hSyn-DIO-mCherry, AAV8-hSyn-DIO-hm3Dq-mCherry, or AAV8-hSyn-DIO-hm4Di-mCherry (Addgene). For $5HT_{2c}$ knockdown experiments, $Htr_{2c}^{lox/lox}$ mice were injected with either AAV8-hSyn-GFP or AAV8-hSyn-Cre-GFP (UNC Vector Core). For fiber photometry experiments measuring calcium in $5HT_{2c}$ neurons, $5HT_{2c}$-Cre mice were injected with AAV8-hSyn-DIO-GCaMP7f (Addgene). For fiber photometry experiments measuring 5HT in $5HT_{2c}$ neurons, $5HT_{2c}$-Cre mice were injected with AAV9-hSyn-DIO-5HT2h (Dr. Yulong Li). For retrograde tracing experiments, wild-type C57BL6/J mice were injected with both Cholera Toxin-B 555 (LHb) and Cholera Toxin-B 647 (BNST) (Thermo Fisher Scientific). Mice were given acute buprenorphine subcutaneously (0.1 mg/kg) on the day of surgery and access to Tylenol in water for 3 days post-op. Mice were allowed to recover in their home cages for at least 4 weeks before the start of experiments.

### Chemogenetic activation of LHb$_{5HT2c}$ or BNST$_{5HT2c}$ neurons

Starting at least 4 weeks after stereotaxic surgery, LHb$_{5HT2c}$ mCherry/hM3Dq and BNST$_{5HT2c}$ mCherry/hM3Dq mice were subjected to a battery of behavioral tests performed in the following order: three chamber sociability, acoustic startle, and open field. One test was performed each day for 3 successive days, and both mCherry and hM3Dq groups received 3 mg/kg Clozapine-N-Oxide (CNO) intraperitoneally (i.p.) 30 min before the start of each behavioral test. One week after the open field test, mice were subjected to three weeks of DiD with no treatment (baseline). On the last day of DiD, both mCherry and hM3Dq mice were given 3 mg/kg CNO i.p. 30 min before a 4 h drinking session. One week following DiD, mice were given one day of 4 h access to either water or a 2% (w/v) sucrose solution with no treatment (baseline). The next day, both mCherry and hM3Dq mice were given 3 mg/kg CNO i.p. 30 min before a 4 h drinking session where both water and a 2% sucrose solution were available. One week after sucrose testing, mice were perfused and viral placement was verified.

### Chemogeneticinhibition of LHb$_{5HT2c}$ or BNST$_{5HT2c}$ neurons

Starting at least 4 weeks after stereotaxic surgery, half of LHb$_{5HT2c}$ mCherry/hm4Di and BNST$_{5HT2c}$ mCherry/hm4Di mice were subjected to three weeks of DiD. 30 min before the last drinking session of the last day of DiD, mice were injected with 3 mg/kg CNO i.p. One week after the last DiD session, mice were subjected to three chamber sociability, acoustic startle, and open field testing. One test was performed each day for 3 consecutive days, and all mCherry and hm4Di mice were treated with 3 mg/kg CNO i.p. 30 min before the start of each behavioral test. One week after these tests, mice were subjected to a sucrose consumption test, again with both mCherry and hm4Di mice receiving acute CNO injections. Mice were then perfused and viral placement was verified.

### Viral-mediated genetic deletion of 5HT$_{2c}$ in LHb and BNST

Starting 4 weeks after stereotaxic surgery, half of LHb$_{5HT2c}$ GFP/Cre and BNST$_{5HT2c}$ GFP/Cre mice were subjected to three weeks of DiD and half continued to have free access to only water. One week following the last DiD session, all mice were subjected to a battery of behavioral tests performed in the following order: three-chamber sociability, acoustic startle, and open field. One test was performed each day for

3 days. One week following the open field test, mice were given one day of 4 h access to either water or a 2% (w/v) sucrose solution. One week after sucrose testing, mice were perfused and viral placement was verified.

### Fiber photometry

**Hardware.** Fiber photometry was performed using a commercially available system from Neurophotometrics, Inc. To record either GCaMP7f or GRAB-5HT signals in LHb$_{5HT2c}$ and BNST$_{5HT2c}$ neurons simultaneously, light from a 470 nm LED was bandpass filtered, collimated, reflected by a dichroic mirror, and focused by a 20× objective into a multi-branch patch cord. Excitation power was adjusted to obtain 75–120 μW of 470 nm light at the tip of the patch cord. Emitted fluorescence was then bandpass filtered and focused on the sensor of a CCD camera and images of the patch cord ROIs corresponding to LHb and BNST were captured at a rate of 40 Hz. 415 nm LED light was also delivered in a similar fashion alternatingly with 470 nm LED light to serve as an isosbestic control channel. To align photometry signals with mouse behaviors, the open-source software Bonsai was used to trigger LEDs simultaneously with behavioral video recording. In a subset of experiments (drinking and startle), TTL pulses were also sent to Bonsai during recordings to identify timestamps of relevant stimuli in real-time through an Arduino-based setup.

**Data collection.** Starting at least 4 weeks after stereotaxic surgery, mice were habituated to patch cords for at least 2 days prior to fiber photometry recordings. Ethanol-naïve male and female mice expressing either GCaMP7f or GRAB-5HT in LHb$_{5HT2c}$ and BNST$_{5HT2c}$ neurons were then subjected to a battery of behavioral tests in the following order: free social interaction, acoustic startle, water drinking, and ethanol drinking. One test was performed each day for 4 successive days, and fiber photometry signals were recorded throughout all behavioral tests. One week following the last behavioral test, half of these mice went through 3 weeks of DiD while the other half continued to only drink water. One week after the last DiD session, all mice were subjected to the same tests in the same order as in the pre-DiD behavioral battery and GCaMP7f or GRAB-5HT signals were recorded and compared between groups.

The free social interaction test was performed in a mouse polycarbonate shoebox cage with dimensions 19 cm × 29 cm × 13 cm (without bedding). Prior to testing, both experimental mice and same-sex, adolescent social targets were habituated to social interaction test cages for 10 min (separately). This habituation was done so that the shoebox cage would be a neutral, but not novel, territory for both mice. For the test, an experimental mouse was placed in the social interaction cage first, then one minute later a social target was added to the cage. The mice were then allowed to freely interact for 10 min and the interaction was recorded with an overhead video camera. The average z-scored $\Delta F/F$ was quantified for the 5 s before and the 5 s after the first social interaction with the juvenile target mouse. Timestamps for the start of the first social interaction were generated manually from videos.

Acoustic startle experiments were performed as described above, except that animals were exposed to only 0 and 120 dB acoustic stimuli rather than 0, 90, 105, and 120 dB acoustic stimuli. The average z-score was quantified for either the 5 s before and the 5 s after each trial stimulus or the 1 s before and 1 s after each trial stimulus and averaged across trial type for each mouse ($n = 10$ blank trials and $n = 10$ 120 dB trials). TTL pulses corresponding to trial stimulus onset timestamps were delivered to Bonsai from the startle apparatus using an Arduino device.

For ethanol and water drinking experiments, an automated Arduino sipper device (Godynyuk et al.[73]) was used to deliver TTL pulses corresponding to lick timestamps to Bonsai. To habituate the mice to the sipper, a dummy sipper device was placed in the home

cage of experimental animals one week prior to testing (mice must drink water from the sipper but the sipper was not hooked up to the Arduino device). On the day of testing, the sipper was placed into the home cage and mice were allowed to freely drink from it until they reached a criterion of at least two isolated drinking bouts more than 10 s apart and lasting at least 5 s each. Mice that did not drink from the sipper within 40 min were excluded from the experiment. For each test, one bottle of either water or 20% (w/v) ethanol was placed in the sipper (separate test days for water vs. ethanol). The average z-score was quantified for the 5 s before and 5 s after the start of each drinking bout and averaged across trials for each liquid.

**Analysis.** Fiber photometry signals were analyzed using a custom MATLAB (MathWorks, Inc.) script. Briefly, 470 nm and 415 nm signals were de-interleaved, background fluorescence was subtracted for each ROI, and the data was low-pass filtered at 2 Hz. Data was then fit to a biexponential curve and the fit was subtracted from the signal to correct for baseline drift. Next, $\Delta F/F$ (%) was calculated for 470 nm and 415 nm signals as 100*(signal-fitted signal)/(fitted signal). For GCaMP7f recordings, the 470 nm signal for the entire recording session was then z-scored and fit using non-negative robust linear regression and the 415 nm signal was fit to the resulting 470 nm signal. The fit 415 nm signal was next subtracted from the z-scored 470 nm signal to yield a motion-corrected recording. For GRAB-5HT recordings, the 470 nm signal was z-scored for the entire recording session without fitting and subtracting the 415 signal, as the 415 nm signal is not an appropriate isosbestic channel for this sensor. We performed additional recordings using AAV-DIO-GFP in 5HT2c-cre mice to ensure a lack of motion artifacts during the behaviors tested (See Fig. S8). Tau (decay constant) of GRAB-5HT signal was computed by fitting a single-phase decay exponential function to the average group trace using GraphPad Prism 9 (GraphPad, Inc.).

## Patch-clamp electrophysiology

Whole-cell patch clamp electrophysiology recordings were obtained from $LHb_{5HT2c}$ and $BNST_{5HT2c}$ neurons using a $5HT_{2c} \times Ai9$ reporter mouse, which expresses tdTomato in $5HT_{2c}$-containing neurons. Recordings from both regions were obtained in the same animals 7–10 days following their last DiD session. Mice were rapidly decapitated under isoflurane anesthesia and brains were quickly removed and immersed in a chilled carbogen (95% $O_2$/5% $CO_2$)-saturated sucrose artificial cerebrospinal fluid (aCSF) cutting solution: 194 mM sucrose, 20 mM NaCl, 4.4 mM KCl, 2 mM $CaCl_2$, 1 mM $MgCl_2$, 1.2 mM $NaH_2PO_4$, 10 mM D-glucose, and 26 mM $NaHCO_3$. Coronal slices containing the LHb or the BNST were prepared on a vibratome at 300 μm and slices were transferred to a holding chamber containing a heated oxygenated aCSF solution: 124 mM NaCl, 4.4 mM KCl, 1 mM $NaH_2PO_4$, 1.2 $MgSO_4$, 10 mM D-glucose, 2 mM $CaCl_2$, and 26 mM $NaHCO_3$. After equilibration for >30 min, slices were placed in a submerged recording chamber superfused with 30–35 °C oxygenated aCSF (2 mL/min). Cells were visualized under a 40× water immersion objective with video-enhanced differential interference contrast, and a 555 LED was used to visualize fluorescently labeled $5HT_{2c}$ neurons. Signals were acquired using an Axon Multiclamp 700B amplifier (Molecular Devices) digitized at 10 kHz and filtered at 3 kHz, and subsequently analyzed in pClamp 10.7 or Easy Electrophysiology. Series resistance ($R_a$) was monitored and cells were excluded from analysis when changes in $R_a$ exceeded 20%. Cells were also excluded from analysis if the current to hold the $V_m$ of the cell at 0 mV in current clamp mode exceeded −100 pA.

Potentials were recorded in current-clamp mode with a potassium gluconate-based intracellular solution: 135 mM K-gluconate, 5 mM NaCl, 2 mM $MgCl_2$. 10 mM HEPES, 0.6 mM EGTA, 4 mM $Na_2ATP$, 0.4 mM $Na_2GTP$, pH 7.3, 289–292 Osm. To hold cells at a similar membrane potential for excitability experiments, $V_m$ was adjusted to

−70 mV by constant current injection. Current injection-evoked action potentials were evaluated by measuring rheobase (minimum current required to evoke an action potential) and number of action potentials fired at linearly increasing current steps (25 pA increments, −100 to 375 pA).

## Immunohistochemistry

To prepare tissue for immunohistochemistry, mice were anesthetized with an overdose of Avertin (1 ml, i.p.) and transcardially perfused with chilled 0.01 M phosphate-buffered saline (PBS) followed by 4% paraformaldehyde (PFA) in PBS. Brains were extracted and post-fixed in 4% PFA for 24 h and then stored in PBS at 4 °C for long-term storage. 45 μm coronal sections were collected using a Leica VT1000S vibratome (Leica Microsystems) and stored in 0.02% Sodium Azide (Sigma Aldrich) in PBS until immunohistochemistry was performed.

To perform immunohistochemistry, tissue was washed for 3 × 10 min in PBS, permeabilized for 30 min in 0.5% Triton-X-100 in PBS, and immersed in blocking solution for one hour (0.1% Triton-X-100 + 10% Normal Donkey Serum in PBS). Next, the tissue was incubated overnight at 4 °C in primary antibody diluted in blocking solution [anti-RFP 1:500 (Rockland, Cat#226-003); anti-5-HT 1:1000 (Immunostar, Cat#200-301-379); anti-cFos 1:1000 (Synaptic Systems, Cat#20079)]. The next day, tissue was washed 4 × 10 min in PBS before being incubated for 2 h in secondary antibody diluted in PBS (all at 1:200) [Donkey anti-mouse Cy2 (Jackson Immunoresearch, Cat#715-125-150), Donkey anti-rabbit Cy3 (Jackson Immunoresearch, Cat#705-165-003), Donkey anti-goat Cy2 (Jackson Immunoresearch, Cat#711-225-152)]. The tissue was then washed 3 × 10 min in PBS, mounted on slides, and allowed to dry overnight before cover slipping with Vecta-Shield Hardset Mounting Medium with DAPI (Vector Laboratories).

## In-situ hybridization

To prepare tissue for in-situ hybridization, mice were anesthetized with isoflurane and rapidly decapitated. Brains were extracted and flash frozen in isopentene (Sigma Aldrich) before being stored at −80 °C until sectioning. 18 μm coronal sections were collected with a Leica CM3050 S cryostat (Leica Microsystems) and mounted directly on slides. ISH was performed to fluorescently label mRNA for mouse serotonin receptor $_{2c}$ (Mm-Htr2c, probe#: 401001), vesicular GABA transporter (Mm-Slc32a1, probe#:319191), and vesicular glutamate transporter 2 (Mm-Slc17a6, probe#: 319171) using the RNAscope Fluorescence Multiplex Assay Kit (Advanced Cell Diagnostics) according to the manufacturer's instructions. Following ISH, the slides were cover slipped with Prolong-Diamond Mounting Medium with DAPI (Thermo Fisher Scientific). For analysis, a minimum of 5 puncta per cell was used as the criteria for a positive cell for any one mRNA marker. 1–4 images were analyzed per animal per experiment.

## Confocal microscopy

All fluorescent images were acquired with a Zeiss 800 upright confocal microscope using Zen Blue software (Carl Zeiss). Validation images of viral expression and optic fiber placement were acquired with a 10× objective, while all other immunohistochemistry and ISH images were acquired using a 20× objective. Images were processed and quantified in FIJI[74].

## Real-time quantitative PCR (qPCR)

To prepare tissue for qPCR, mice were anesthetized with isoflurane and rapidly decapitated. Brains were extracted and immediately placed on ice in PBS buffer, then placed in a brain cutting block and sectioned into 1 mm thick slices. Brain punches containing the LHb (one punch) or the BNST (bilateral punches) were collected from 1 mm sections using a 1 mm diameter tissue micro-punch and immediately flash-frozen in a tube block chilled with dry ice. Tissue punches were kept in a −80 °C freezer until processing. RNA was extracted from brain

tissue using the RNAeasy Kit (Qiagen) per the manufacturer's instructions and eluted in water. Following extraction, RNA concentration and quality was assessed using a NanoDrop Spectrophotometer (Thermo Fisher). RNA concentrations were then normalized to 1 ng/μL. Reverse transcription of cDNA from total RNA was performed using Super-Script IV VILO Master Mix with ezDNase according to the manufacturer's instructions (Thermo Fisher). For each qPCR reaction, 1 ng of cDNA was combined with 10 μl of Taqman Advanced Master Mix (Thermo Fisher), 0.5 μl each of Taqman mouse *Htr2c* (cat. 401001) and *GAPDH* primers (cat. 4331182), and topped off with water to total 20 μl for each reaction. For each mouse, three biological replicates were included for each brain region. *GAPDH* and *Htr2c* expression was assessed within single replicates using a multiplex technique with FAM (*Htr2c*) and VIC (*GAPDH*) dyes. Real-time qPCR was performed using the QuantStudio3 System (Thermo Fisher) and analyzed using the $2^{-\Delta\Delta CT}$ method with *GAPDH* as a housekeeping gene for normalization of *Htr2c*[75].

## Statistics

Single-variable comparisons between two groups were made using paired or unpaired two-tailed *t* tests. Group comparisons were made using one-way ANOVA, two-way ANOVA, or two-way mixed-model ANOVA (depending on the number of independent and within-subject variables). Following significant interactions or main effects, post-hoc pairwise *t* tests were performed using Holm-Sidak's test to control for multiple comparisons. All data are expressed as mean ± standard error of the mean (SEM), with significance defined as $p < 0.05$ unless otherwise noted. All data were analyzed with GraphPad Prism 9 (GraphPad Software). A table of full statistical information for this manuscript is included as a Supplementary Table (Flanigan et al. 2023 Statistics).

## Excluded data

Data points were only excluded from these analyses for the following reasons: missed targeting of viral injections or optic fiber placements, clogged or spilled water/sucrose/alcohol bottles, malfunctions in fiber photometry hardware affecting the quality of data collected, data points that were statistically significant outliers (as determined using Grubb's test, used only once per dataset to identify single outliers), or cells in electrophysiology experiments that did not meet inclusion criteria for health (see electrophysiology section above for these criteria).

## Reporting summary

Further information on research design is available in the Nature Portfolio Reporting Summary linked to this article.

## Data availability

Source data are provided with this paper as a Source Data file (Source Data Flanigan et al. 2022). Raw Fiber Photometry data and associated videos are promptly available upon request, but are not immediately available to download due to file size. Please email the corresponding author Dr. Thomas Kash (Thomas_kash@med.unc.edu) to obtain these data. Source data are provided with this paper.

## Code availability

Custom MATLAB scripts developed to process raw fiber photometry data can be found in a GitHub repository (https://github.com/meghanflanigan/Flanigan_et_al_2023).

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

## Acknowledgements

This work was supported by grants from the National Institutes of Health's (NIH) National Institute of Alcohol Abuse and Alcoholism (NIAAA) (M.F.: T32 AA007573-21; T.K.: R01 AA019454-12). Figures for this manuscript were created using BioRender.com. We also thank Dr. Dipanwita Pati for their feedback over the course of this project.

## Author contributions

M.E.F. and T.L.K. conceptualized experiments. M.E.F. performed behavior, electrophysiology, fiber photometry, chemogenetics, histology, qPCR, and analyzed data. L.H. assisted with behavioral experiments. H.L.H., M.M.P., and S.D. assisted with drinking experiments. K.M.B. performed in-situ hybridization experiments and qPCR. M.C. analyzed in-situ hybridization images. O.J.H. assisted with fiber photometry data collection and analysis. M.E.F. and T.L.K. wrote the paper with editing contributions from all authors.

## Competing interests

The authors declare no competing interests.
