## [Peer Review File · Nature Communications]

Sex-specific regulation of binge-like alcohol consumption, social, and arousal behaviors in mice by 2 subcortical serotonin 5HT2c receptor-containing neuronsREVIEWER COMMENTS

Reviewer #1 (Remarks to the Author):

Summary- Flanigan and colleagues present a manuscript that rigorously characterizes the effects of binge alcohol drinking, and social/affective behaviors on 5HT signaling in two brain regions- the lateral habenula and BNST. The authors identify numerous interesting sex- and region-specific differences in 5HT signaling on social interaction, acoustic startle, binge drinking, and other behaviors. For example, in vivo calcium photometry recordings in the BNST suggests increased activity in male mice during social interaction tasks and acoustic startle tests, while this sex difference is not observed in the LHb. However, in both brain regions, male mice show increased responsiveness (larger decrease in calcium signal) during ethanol drinking bouts, outlining potential common circuitry driving sex differences in alcohol drinking. Interestingly, these results were not observed when a specific 5HT sensor was used, suggesting release of 5HT onto 5HT2C- expressing cells is not a reason for their observation. The authors use a number of exciting techniques including in vivo recording of 5HT release with the GRAB-5HT fluorescence sensor, region specific modulation of 5HT with 5HT2C cre mice, anatomical and neurochemical circuit tracing, acute slice electrophysiology, and sound behavioral testing. While the rigorous approach and robustness of the manuscript is a strength, the many factors being compared throughout the manuscript make it difficult to follow and at times it seems rather descriptive. The authors make a compelling case for focusing on 5HT in the LHb and BNST and the potential importance of these regions as they relate to sex differences. However, in its current form, it is difficult to ascertain the overall hypothesis being tested. It seems like two parallel stories are being told on the LHb and BNST. A critical missing piece is some sort of converging experiment convincing the reader that the DRN-LHb and DRN-BNST are essential inputs regulating these behaviors, and not just two examples of inputs involved in binge drinking. Along these lines, the sex differences are very interesting but I'm not sure what the overall takeaway should be from these observations. Perhaps a model figure or more clearly stating the rationale for why each experiment is being performed as it relates to the overall story would help the reader. In all, the topic is highly relevant and will be of interest to the broad scientific community, the experiments are well designed, and the techniques and models are novel, innovative, and optimal for addressing the research questions. Creating a more structured, succinct story that either focuses on one brain region, or experimentally implicates the collective role of 5HT signaling in both brain regions in regulating sex-specific effects on social, affective, and binge drinking behavior would vastly improve this manuscript. More specific comments and concerns are below.

1) The drinking levels are quite variable across experiments. Why are there significant sex differences in figure 4B for both 2h and 4h sessions, but no differences in 4P nor the drinking data presented in figures 5 and 6? It looks like the female mice used in 4A-L drank significantly more than in the other experiments, leading to the effect.

2) For the electrophysiology experiments, what is the rationale for restricting recordings to the vBNST but then collapsing data from medial and lateral LHb? Are there differences in the dBNST? This is particularly important given that the DREADD expression in Figure 3 is both dorsal and ventral.

3) How was the correlation between consumption and RMP determined? Is this the average RMP for x number of cells per animal? If so this should be indicated on the graph and in the text.

a. Given the sex difference observed in 5K, how do the authors justify collapsing the data in 5L?

b. 5L has a male data point between 4 and 5 g/kg but there is no male that drank above 4 according to 5B. I'm assuming this is an error and that data point is female?

4) The RMPs for cells in both regions seems rather high. Is this a known characteristic of 5HT2c cells? I'm particularly confused how the average AP threshold for the BNST is lower than the RMP. Are these cells tonically active? But based on the rheobase in 5M, it looks like only 1 male cell is. How do the authors explain this? Some discussion on this would be informative.

5) For figure 6, I'm not sure randomly assigning mice to a group is the best strategy given the range of responses in all behaviors in figure 1. In nearly every experiment there were responses that were in the opposite directions of the overall response, which is not necessarily uncommon but should be accounted for. For example, some mice had negative average z-score on the acoustic startle, while the average is positive. Did random assignment group these negative mice into the water group? And are these the same mice with negative average z-scores in P? Either way, a potential strength of this experiment is the within-subject design, and it would be interesting if the authors presented more data comparing before/after drinking.

a. In this vein, the water mice had exposure to alcohol during the initial test, so these mice are not alcohol naïve as stated on page 9 line 13 and in figure 6a.

b. Could repeated exposure to the behaviors impact these results?

6) For the photometry experiments, what is the average z-score presented? Is this a specific time after the onset of the behavior or the entire 10 seconds? It would be interesting to look at an additional measure like area under the curve. For instance, figure 10 looks like a large male/female difference but the average z-score is very similar.

Reviewer #2 (Remarks to the Author):

The present manuscript examines the contribution of LHb and BNST 5HT2c neuron activity and 5HT2c receptor within these areas to a battery of behavioral tests aimed at social behavior and arousal outside and inside the context of binge alcohol drinking. The premise is that binge drinking is associated with impairments in social behavior and arousal, which could promote future consumption. Understanding the underlying neural mechanisms supporting such negative effects is warranted. Many approaches were taken, including dual fiber photometry (LHb and BNST) of calcium or a 5HT sensor were performed, along with some anatomical work, ex vivo characterization of basic properties of neurons within this area following water or DID exposure, and deletion of 5HT2c from these specified populations. Overall, the authors present a description of how these different populations may be affected by DID and sex differences, and how that may modulate social behaviors, arousal, and binge drinking. Some of the findings are novel and important; however, the broader scope of the paper is quite descriptive with the main findings not fully supported by the type of analyses performed, or from the associated statistics. In reviewing this manuscript, it was hard to get past these problems, and there did not appear to be a significant advancement in our understanding, thus my overall enthusiasm is low.

Main concerns;

There is no information about the microstructure of behaviors emitted, bout statistics, licks, etc.,. This produces problems for comparisons between groups (e.g. DID and water mice), as average and AUC analyses are used to test statistical significance, yet the underlying behavior is varied. It is not at all clear that these differences reflect differences in the behavior emitted or activity recruitment (as suggested). Time locked or interpolated analyses to the behavior emitted are necessary to make any conclusions from these data types. The same is said for all other behaviors when making comparisons between two groups (male/female, water, DID).

Through the figures, there are many statistically unsupported analyses and conclusions. Often, there is no significant interaction but post hocs are still performed. There can be pre-planned reasons to investigate a main effect (magnitude effects for example), but in the present manuscript, post hocs are not limited to main effects and are made across factors when not warranted. For example, in Figures 3 and 4 this pertains to 3G, 3H, 3Q, 3S, 4H, 4I, 4J, 4N, 4S, but is seen throughout the manuscript and in supplemental material.

Experiments used the DID drinking in the dark procedure, which produces relevant BECs only during the 4 h, and very little ethanol preference (different from 0.5)- hence the claim of rewarding is a bit weak (although one could argue alcohol and water were equally rewarding, but that interpretation begs the question of alcohol as a reward in the drinking model). However, Figure 7 with the knockdown of 5HT2c only shows the 2 hr results- which seems a bit misleading; where is the 4 hr DID data where they are drinking pharmacologically relevant levels.

Additional concerns;

Most photometry traces throughout the manuscript are z-scored, but it is not stated what it is z-scored to, with no explanation as to why that baseline. This is extremely important and should be carefully thought out as most experiments do not have a trial structure.

Not clear if traces are per animal or per behavioral onset (collapsed across animals). While fine with the latter, what evidence is there that this is seen across animals instead of a single or few animals driving effect?

For anatomical characterization, very little information is given about how percentages were normalized based on infection rate and spread. It is done by percentage of neurons in the DRN, but differences in injection success and uptake in BNST and LHb do not appear to be accounted for. Thus, making claims about male/female differences appears to be premature. It is a tricky thing to do, but nevertheless should limit interpretation.

Response to Reviewers- Flanigan et al.

We thank the reviewers for their helpful comments and believe the changes they suggested have much improved our manuscript. We believe we have addressed all of the reviewer concerns in our revision, and we specifically address reviewer comments below.

Reviewer 1:

Summary- Flanigan and colleagues present a manuscript that rigorously characterizes the effects of binge alcohol drinking, and social/affective behaviors on 5HT signaling in two brain regions- the lateral habenula and BNST. The authors identify numerous interesting sex- and region-specific differences in 5HT signaling on social interaction, acoustic startle, binge drinking, and other behaviors. For example, in vivo calcium photometry recordings in the BNST suggests increased activity in male mice during social interaction tasks and acoustic startle tests, while this sex difference is not observed in the LHb. However, in both brain regions, male mice show increased responsiveness (larger decrease in calcium signal) during ethanol drinking bouts, outlining potential common circuitry driving sex differences in alcohol drinking. Interestingly, these results were not observed when a specific 5HT sensor was used, suggesting release of 5HT onto 5HT2C- expressing cells is not a reason for their observation. The authors use a number of exciting techniques including in vivo recording of 5HT release with the GRAB-5HT fluorescence sensor, region specific modulation of 5HT with 5HT2C cre mice, anatomical and neurochemical circuit tracing, acute slice electrophysiology, and sound behavioral testing. While the rigorous approach and robustness of the manuscript is a strength, the many factors being compared throughout the manuscript make it difficult to follow and at times it seems rather descriptive. The authors make a compelling case for focusing on 5HT in the LHb and BNST and the potential importance of these regions as they relate to sex differences. However, in its current form, it is difficult to ascertain the overall hypothesis being tested. It seems like two parallel stories are being told on the LHb and BNST. A critical missing piece is some sort of converging experiment convincing the reader that the DRN-LHb and DRN-BNST are essential inputs regulating these behaviors, and not just two examples of inputs involved in binge drinking. Along these lines, the sex differences are very interesting but I'm not sure what the overall takeaway should be from these observations. Perhaps a model figure or more clearly stating the rationale for why each experiment is being performed as it relates to the overall story would help the reader. In all, the topic is highly relevant and will be of interest to the broad scientific community, the experiments are well designed, and the techniques and models are novel, innovative, and optimal for addressing the research questions. Creating a more structured, succinct story that either focuses on one brain region, or experimentally implicates the collective role of 5HT signaling in both brain regions in regulating sex-specific effects on social, affective, and binge drinking behavior would vastly improve this manuscript. More specific comments and concerns are below.

We thank the reviewer for their positive comments and their helpful suggestions. We have made a number of changes and additions to the manuscript that we believe have improved it substantially. Major changes and additions include:

- A. Providing clearer rationales and hypotheses for experiments (changes shown in **blue** in manuscript)
- B. Adding a summary figure of findings
- C. Adding heatmaps of individual animal responses in fiber photometry experiments
- D. Adding additional functional experiments to reduce descriptive nature of manuscript. This was a valid point in the first submission. While the manuscript was under revision, we completed an experiment using a chemogenetic approach to determine if activity in either the BNST or LHb contributed to ongoing behavioral deficits. Much to our surprise, activation of Gi signaling in the LHb reversed the changes in behavior. We think that this data adds important mechanistic insight to the work.
 - a. Chemogenetic inhibition of LHb_{5HT2c} and BNST_{5HT2c} in alcohol-exposed and alcohol-naïve mice
 - b. Chemogenetic inhibition of LHb_{5HT2c} fully normalized social disturbances induced by alcohol in females and startle disturbances induced by alcohol in males.
 - c. Chemogenetic inhibition of BNST_{5HT2c} did **not** normalize social behavior in females, indicating that LHb_{5HT2c} is the critical population regulating alcohol-induced social recognition deficits

- d. Chemogenetic inhibition of BNST_{5HT2c} also normalized startle behavior, indicating that BNST and LHb populations both contribute to alcohol-potentiated startle
- e. Chemogenetic inhibition of LHb_{5HT2c} robustly increased alcohol consumption in females. Thus, LHb_{5HT2c} neurons appear to regulate drinking in females. This is consistent with our photometry findings showing that following DiD, females show greater decreases in LHb_{5HT2c} calcium activity during alcohol consumption. Our finding that chemogenetic inhibition of LHb_{5HT2c} did not alter drinking in males is supported by previous work showing that pan-LHb chemogenetic inhibition in male rats does not alter drinking (Nentwig et al. 2022).

1) The drinking levels are quite variable across experiments. Why are there significant sex differences in figure 4B for both 2h and 4h sessions, but no differences in 4P nor the drinking data presented in figures 5 and 6? It looks like the female mice used in 4A-L drank significantly more than in the other experiments, leading to the effect.

The reviewer raises a good point. A significant challenge in alcohol research is achieving high levels of voluntary alcohol intake in transgenic mouse lines. The mice used in Figure 4 were wild-type C57BL/6J mice, which are known to consume high levels of alcohol. While all of our transgenic lines (5HT2c-cre, 5HT2c-flox, 5HT2c-Ai9) have been backcrossed on to a C57BL/6J background, they do indeed consume less than wild-types, and at times females do not consume more than males. However, despite our transgenic mice drinking less than wild-types, we will note that transgenic mice still displayed similar alcohol-induced behavioral changes observed in wild-types. For example, in Figs. 7 and 8, female mice reliably showed social recognition deficits following DiD even though their alcohol intake was markedly lower than wild-types. In addition, we will note that the drinking levels we observed in our transgenic mice are in fact pharmacologically relevant despite being lower than wild types. In our studies, we observed on average 3 g/kg intake in 2h of DiD, which we know from previous experiments produces BECs with an average of ~127 mg/dl in transgenic mice on a C57BL/6J background (**see below, unpublished data from Kash Lab**). The consistent finding that mice tend to front-load their alcohol consumption to the beginning of drinking sessions (Wilcox et al. 2014) may explain why our 4h BECs in Figure 4 appear similar to the 2h BECs we observe in animals with lower overall intake.

2) For the electrophysiology experiments, what is the rationale for restricting recordings to the vBNST but then collapsing data from medial and lateral LHb? Are there differences in the dBNST? This is particularly important given that the DREADD expression in Figure 3 is both dorsal and ventral.

The reviewer raises an important point here. Previous data from our laboratory shows that of the BNST sub-regions, only vBNST neurons display increased 5HT2c-dependent excitability following alcohol consumption (Marcinkiewicz et al. 2015). In a set of unpublished experiments, we also observed that alcohol vapor exposure increases Fos activation of 5HT2c neurons in the vBNST, but not in the dBNST. As a result, we performed our electrophysiology experiments specifically in the vBNST sub-region. Unfortunately, restricting viral injections to the vBNST alone is very difficult in practice. While we did get vBNST viral expression in all of our animals, we did also see some viral spread to the dBNST. We recognize this as a potential caveat, as there are notable differences between the circuitry of dBNST versus vBNST. We have made note of this caveat in the discussion of our revised manuscript.

In the case of the LHb, there was no pre-existing data on sub-regional differences in the effects of alcohol on neuronal physiology; therefore, we recorded from both medial LHb and lateral LHb neurons. However, upon inspecting our LHb data, we observed a sub-region specific difference in females. We now include in the manuscript specific data from the medial LHb in females (Fig. 5), where we observed an increase in rheobse.

3) How was the correlation between consumption and RMP determined? Is this the average RMP for x number of cells per animal? If so this should be indicated on the graph and in the text.

a. Given the sex difference observed in 5K, how do the authors justify collapsing the data in 5L?

b. 5L has a male data point between 4 and 5 g/kg but there is no male that drank above 4 according to 5B. I'm assuming this is an error and that data point is female?

We apologize for not clarifying how the RMP vs. consumption correlation was performed. Yes, we averaged the RMP for each animal (1-4 cells/mouse) and associated this average with their average intake over the 3 weeks of DiD. We agree that given the sex differences in RMP, it may not have been appropriate to combine males and females together for the correlation, lest there be a spurious correlation due to inherent differences in drinking and RMPs. However, since we observed an intermixed range of consumption and RMPs in males and females and our females did not drink more than our males in this experiment, we plotted them on the same scatterplot. Unfortunately, splitting the RMP correlations with consumption into separate male and female groups yields datasets that are underpowered to pick up any significant correlations. Thus, we have removed the correlations from the manuscript.

We apologize for the error in the scatterplot, that mouse drinking over 4 g/kg was in fact a female animal.

4) The RMPs for cells in both regions seems rather high. Is this a known characteristic of 5HT2c cells? I'm particularly confused how the average AP threshold for the BNST is lower than the RMP. Are these cells tonically active? But based on the rheobase in 5M, it looks like only 1 male cell is. How do the authors explain this? Some discussion on this would be informative.

Previous studies suggest that BNST CRF neurons, which overlap with BNST 5HT2c neurons, are occasionally in depolarization block, particularly in females (Levine et al. 2021). Thus, their RMPs are high but the cells are silent at this RMP. This may explain why the average AP threshold for the BNST is lower than the RMP, as the AP threshold was performed while holding the cells at -70 mV. **Below** we show a breakdown of cell types in males and females for water and DiD groups (tonic firing, intermittent firing/bursting, silent, in depolarization block=silent at greater than -40 mV). We have added this characterization of cell properties to Fig 5 so that the readers may better interpret our findings.

5) For figure 6, I'm not sure randomly assigning mice to a group is the best strategy given the range of responses in all behaviors in figure 1. In nearly every experiment there were responses that were in the opposite directions of the overall response, which is not necessarily uncommon but should be accounted for. For example, some mice had negative average z-score on the acoustic startle, while the average is positive. Did random assignment group these negative mice into the water group? And are these the same mice with negative average z-scores in P? Either way, a potential strength of this experiment is the within-subject design, and it would be interesting if the authors presented more data comparing before/after drinking.

a. In this vein, the water mice had exposure to alcohol during the initial test, so these mice are not alcohol naïve as stated on page 9 line 13 and in figure 6a.

b. Could repeated exposure to the behaviors impact these results?

The points about random assignment are well taken—indeed there are some mice that show a negative average z-score, but this is likely due to the 5s time window used for analysis post startle delivery (the startle spikes are only 1-2 seconds long, max). If looking at a smaller time window, or looking at peak values only, all mice show positive spikes in response to the startle stimuli. We chose 5s as the analysis window here because 5s had been used for all other tasks and wanted to remain consistent in our analyses. However, when considering other analysis windows (1-3 s) our overall conclusions did not change. We have clear data showing that if we split the groups from Fig 1 into mice that eventually got DiD treatment and mice that eventually got water treatment, there are no pre-existing differences between the groups in either sex (**see examples C and D below**). In addition, the mice that had small responses in one behavior were not necessarily the same mice that showed small responses in another behavior. For example, for BNST recordings in males, an individual mouse that had one of the smallest startle GCaMP signals showed the largest GCaMP social signal (**see E below, orange arrow denotes individual male we are referring to**). Hence, we believe that the differences we observe in Fig 6 are truly in response to DiD exposure.

- A. We apologize for stating that the water mice were alcohol-naïve, as you are correct, they were exposed very briefly to alcohol in the pre-DiD test. So while they did not have binge drinking experience, though they had tasted alcohol briefly before.
- B. Yes, it is possible that repeated exposure to behaviors could have impacted the results, but in that case we would expect things to be changing in both the DiD mice and the water mice, as both groups were exposed to each set of behaviors twice. The relatively long time period (1 month) between tests also reduces the likelihood that the post-DiD behaviors are impacted by pre-DiD exposure. However, we now bring this up as a potential caveat in the discussion. Though we did not observe changes between testing sessions in control animals, we did observe acute experience-dependent changes in neural activity within the first alcohol exposure session. Below, we display additional data comparing first vs. last alcohol drinking bouts in previously alcohol-naïve mice, where we observed a decrease in LHB_{5HT2c} signal in females in the last bout compared to the first.

C. Female DiD and Water groups do not show pre-existing differences in GCaMP signals upon drinking alcohol (below).

D. Male DiD and Water groups do not show pre-existing differences in GCaMP signals in the acoustic startle test (below).

GCaMP7f in BNST, pre-DiD males startle GCaMP7f in LHb, pre-DiD males startle

E. Individual animals that show small responses to one stimulus do not simply show small responses to all stimuli, orange arrow denotes one individual male animal across tests.

GCaMP7f in BNST, Startle GCaMP7f in BNST, Social

6) For the photometry experiments, what is the average z-score presented? Is this a specific time after the onset of the behavior or the entire 10 seconds? It would be interesting to look at an additional measure like area under the curve. For instance, figure 10 looks like a large male/female difference but the average z-score is very similar.

As we state in the methods and the figure legends, the time window we used for average z-score in these experiments was the average in the 5s immediately following the behavior/stimulus delivery. We also looked at AUC for all experiments and the results did not differ from the averages in terms of significant differences. We have also run this analysis using smaller average time windows (1-3s) and/or max peak values and also did not pick up any additional differences. Hence, we showed only the 5s average data in the manuscript.

Reviewer 2:

Summary: The present manuscript examines the contribution of LHb and BNST 5HT2c neuron activity and 5HT2c receptor within these areas to a battery of behavioral tests aimed at social behavior and arousal outside and inside the context of binge alcohol drinking. The premise is that binge drinking is associated with impairments in social behavior and arousal, which could promote future consumption. Understanding the underlying neural mechanisms supporting such negative effects is warranted. Many approaches were taken, including dual fiber photometry (LHb and BNST) of calcium or a 5HT sensor were performed, along with some anatomical work, ex vivo characterization of basic properties of neurons within

this area following water or DID exposure, and deletion of 5HT2c from these specified populations. Overall, the authors present a description of how these different populations may be affected by DID and sex differences, and how that may modulate social behaviors, arousal, and binge drinking.

Some of the findings are novel and important; however, the broader scope of the paper is quite descriptive with the main findings not fully supported by the type of analyses performed, or from the associated statistics. In reviewing this manuscript, it was hard to get past these problems, and there did not appear to be a significant advancement in our understanding, thus my overall enthusiasm is low.

We thank the reviewer for their time and their suggestions. We respectfully disagree with the point regarding the main findings and statistics not supporting the conclusions we reach in the manuscript. The specific statistical issues the reviewer seems to be referring to involve the comparisons we made in post-hoc tests following ANOVAs. While we agree with the reviewer that comparisons between males and females are inappropriate due to their different levels of drinking, we do not think removing these comparisons in any way challenges any of the main conclusions of the paper. Indeed, we hesitated to make direct comparisons between males and females in our discussion, but rather spoke about the differential effects of alcohol on males and females separately.

To make our manuscript less descriptive, we have added additional functional experiments in which we investigated the effects of chemogenetic inhibition of LHB_{5HT2c} and BNST_{5HT2c} on binge drinking, social behavior, and startle behavior in both sexes. Unlike 5HT2c deletion, which only partly normalized alcohol-induced social and startle dysregulation, chemogenetic inhibition of these neurons, particularly in the LHB, fully normalized behavior in alcohol-exposed animals. These findings have strengthened the overall conclusions of our paper, identifying the activity of LHB_{5HT2c} as a critical regulator of alcohol-induced behavioral dysfunction.

1) *There is no information about the microstructure of behaviors emitted, bout statistics, licks, etc.. This produces problems for comparisons between groups (e.g. DID and water mice), as average and AUC analyses are used to test statistical significance, yet the underlying behavior is varied. It is not at all clear that these differences reflect differences in the behavior emitted or activity recruitment (as suggested). Time locked or interpolated analyses to the behavior emitted are necessary to make any conclusions from these data types. The same is said for all other behaviors when making comparisons between two groups (male/female, water, DID).*

If we are understanding the reviewer correctly, they are concerned that different patterns of behavior during the photometry recordings may be the reason underlying differences in signals. We believe there are several reasons for why this is not an issue in our experiments. First, for social behavior testing, the behavior emitted was the same in each mouse. We aligned the photometry signals to the first moment the experimental mouse investigated the novel social target, which is an event that lasts at **least** as long as the 5s analysis window we used for quantification. Hence, all mice were actively engaging in prosocial investigation during the analysis window. For startle behavior testing, all mice engaged in some magnitude of startle response that was above that of a blank trial, and all mice received the same number of startle stimuli in the same (random) order. The photometry signals were aligned to the moment the startle stimulus was delivered for each trial. It is indeed true that the magnitude of startle response was not equal between mice (**see male startle behavior data below**), but the length of time of the response was not different between mice—hence, all mice were performing the same behavior at the same time. For drinking behavior experiments, we understand why the reviewer may be concerned about how differences in voluntary drinking behavior could be contributing to the differences in signals observed. However, we dealt with this potential issue by setting a minimum criteria for each bout that would be included in the analysis such that all mice were actively licking alcohol for the entirety of the 5s analysis window. As we state in the methods, our minimum criteria was at least two isolated drinking bouts more than 10s apart and lasting at **least** 5s. Hence, because all animals were engaging in the same behaviors for the entirety of the analysis windows for all photometry experiments, we believe it is perfectly appropriate to directly compare neural signals between groups using a common analysis time window.

- 2) *Through the figures, there are many statistically unsupported analyses and conclusions. Often, there is no significant interaction but post hoc are still performed. There can be pre-planned reasons to investigate a main effect (magnitude effects for example), but in the present manuscript, post hoc are not limited to main effects and are made across factors when not warranted. For example, in Figures 3 and 4 this pertains to 3G, 3H, 3Q, 3S, 4H, 4I, 4J, 4N, 4S, but is seen throughout the manuscript and in supplemental material.*

It is our understanding that post-hoc comparisons may be done following **either** significant main effects or significant interactions, but that the specific post-hoc comparisons being performed should be experimentally appropriate and, indeed, pre-planned. We **only** performed post-hoc comparisons when main effects or interactions were significant. However, we agree that it was inappropriate to perform comparisons between males and females in Figures 3 and 4 considering their different degrees of alcohol consumption. We have now updated the statistics in these figures and throughout the manuscript so that post-hoc comparisons between water and DiD groups are performed within each sex and not across sexes. This change does not impact any of our conclusions, as we simply sought to look at qualitative differences in the effects of alcohol/manipulations on males versus females.

- 3) *Experiments used the DID drinking in the dark procedure, which produces relevant BECs only during the 4 h, and very little ethanol preference (different from 0.5)- hence the claim of rewarding is a bit weak (although one could argue alcohol and water were equally rewarding, but that interpretation begs the question of alcohol as a reward in the drinking model). However, Figure 7 with the knockdown of 5HT2c only shows the 2 hr results- which seems a bit misleading; where is the 4 hr DID data where they are drinking pharmacologically relevant levels.*

As we mention in our response to Reviewer 1, mice **do** achieve pharmacologically relevant, intoxicating levels of alcohol intake in 2h of consumption in the DiD paradigm. In our studies, we observed on average 3 g/kg intake in 2h of DiD, which we know from previous experiments produces BECs with an average of ~127 mg/dl in transgenic mice on a C57BL6/J background (**see below, unpublished data from Kash Lab**). The consistent finding that mice tend to front-load their alcohol consumption to the beginning of drinking sessions (Wilcox et al. 2014) may explain why our 4h BECs in Figure 4 appear similar to the 2h BECs we observe in animals with lower total intake. While preference on average was indeed between 0.5 and 0.6, the purpose of this study was not to investigate the rewarding properties of alcohol, but to investigate the effects of alcohol on social and arousal behaviors. We did not include the 4h data in Figure 7 for the sake of space, but the findings remain the same regardless of whether we consider 2h or 4h consumption. We have now added in the 4h data for Figures 7 and 8.

- 4) *Most photometry traces throughout the manuscript are z-scored, but it is not stated what it is z-scored to, with no explanation as to why that baseline. This is extremely important and should be carefully thought out as most experiments do not have a trial structure.*

We apologize for not communicating our photometry analysis more clearly in our manuscript. For all experiments, the photometry signals were z-scored to the entire recording session, not a pre-defined baseline period. Indeed, we did this because some of the photometry experiments did not have a trial structure. This method of z-scoring is indeed the most **conservative** method.

- 5) *Not clear if traces are per animal or per behavioral onset (collapsed across animals). While fine with the latter, what evidence is there that this is seen across animals instead of a single or few animals driving effect?*

We apologize for not communicating this more clearly in the original manuscript. For all photometry traces, trials were averaged for each mouse and then that average was plotted. We did this because for drinking experiments there were not always equal numbers of bouts being compared—plotting individual bouts could indeed skew the results in favor of a single mouse with the most bouts. Hence, we do not believe the signals observed are being driven by single animals. To illustrate the responses of individual animals more clearly, we now include two additional supplemental figures in our manuscript (Figures S6 and S8). These figures include heatmaps for calcium and 5-HT signals during each of the behavioral tasks, with the average of all trials for each individual animal displayed as a distinct horizontal line in the heatmap (**shown below as well**).

GCaMP7f in LHb_{5HT2c}

GCaMP7f in BNST_{5HT2c}

6) For anatomical characterization, very little information is given about how percentages were normalized based on infection rate and spread. It is done by percentage of neurons in the DRN, but differences in injection success and uptake in BNST and LHB do not appear to be accounted for. Thus, making claims about male/female differences appears to be premature. It is a tricky thing to do, but nevertheless should limit interpretation.

We apologize for not adequately explaining our analysis in the original manuscript. Indeed, quantitative comparisons in anatomical tracing studies can be tricky to interpret, and thus we hesitated to perform any kind of analysis of the relative strength of innervation of DRN-BNST and DRN-LHB in males and females. Instead, we calculated percentages of neurons in the DRN, but unfortunately did not normalize to the relative infection area in the LHB/BNST. Due to this limitation, we have removed the direct statistical comparison between males and females in our tracing figure (Figure 2) and now simply report the values obtained in males and females.

REVIEWER COMMENTS

Reviewer #1 (Remarks to the Author):

I thank the authors for thoroughly responding to my comments. The revised manuscript is much improved in my opinion. The addition of the DREADD experiments in figure 8 certainly addresses a major concern with the descriptive nature of the first submission. However, there are still a few areas of concern that should be addressed.

1) I would like to push back slightly on the responses to my comments on the fiber photometry experiments in figures 1 and 6. Choosing 5 seconds still seems very arbitrary considering that the signal does not appear to return to baseline until well after 5 seconds in the case of social interaction, water drinking, and ethanol drinking. This is even more interesting considering the stability of the data prior to the bout now that it is clearer that the Z-score is not calculated within this specific window.

Of equal concern is the acoustic startle, which returns to baseline much quicker than 5 seconds. To the eye looks like there are several differences that are going undetected with the current analyses.

a) Based on how the data are presented in figure 1O, looks to me to be a very large male/female difference in peak response (more than double), but the average over 5 seconds is not different.

b) GRAB-5HT data in 1AA and II- similar peak but male activity stays elevated much longer than female

c) Figure 6 M vs O and Q vs S- male is significant but this is driven by an increase well after the stimulus has ended, whereas the peak z-score is similar in males and females, if anything both are trending towards an increase

d) Figure 6 GG- the DiD response is quite different than the water and again an effect at the time of the stimulus appears to be washed out by a negative bump well after the stimulus.

I fully appreciate the complexity of the analysis and interpretation of the results and wanting to perform uniform analyses across experiments, but in this case, uniformity detracts from potential interesting sex- and DiD-dependent differences. The authors mention in the response (but not the main text) that maximum peak and AUC data are negative across the board, but based on the visual representation of average traces, it would be informative to include these negative data and other behavior specific measurements to address my, and likely other readers, confusion with the data as presented. This could also be very informative for other research attempting to follow up on these interesting results.

2) I appreciate the authors pointing out the caveats with their work, however I am still concerned with the dorsal vs ventral BNST virus expression. Specifically, the authors state on page 19 line 17-20 that "...in some cases did observe spread to the dBNST". If the spread into dorsal BNST is only in some cases this raises some new concerns. The authors should simply remove the cases where virus spread into the dBNST. This could also clarify some of the difficulties with interpreting the new DREADD data. Alternatively, some sort of control electrophysiology experiment could be performed to convince the reader this spread is not driving the results of the DREADD experiments.

a) I'm similarly concerned that the representative image of fiber placement for BNST GCaMP (Figure 1B) is in the dorsal BNST while the GRAB-5HT image is in the ventral BNST (Figure 1V). The BNST sub-region discrepancies throughout the manuscript should be addressed.

Minor- the authors state that 415 was the isosbestic wavelength used for the experiments but the addition on page 4 line 26 says 405. Please clarify.

Reviewer #2 (Remarks to the Author):

The addition of mechanistic data (the chemogenetic inhibition experiments) improves the story and excitement for this manuscript. The novel take home from this work is that the LHb population is important for disturbances to social behavior following binge-like drinking in female mice. However, this appears not to be due to 5HT_{2c}, but instead in relation to overactivation of this population. Overall, the paper still reads a bit descriptive and slightly confusing in places, but is improved in flow and now more focused on the main findings.

In regards to the new data, they point to a novel role for the LHb5HT_{2c} population in social deficits and increased alcohol consumption in females, but not males, while chemogenetic inhibition of the BNST 5HT_{2c} population normalized startle behaviors.

My apologies for not understanding the drinking bout duration criteria. For photometry traces, a bar above the graph indicating ongoing behavior would be useful for many reasons, first to avoid confusion about whether animals are behaving differently, but also in regards to the dynamics that evolve across the period. While the behaviors are unlikely to be exactly the same, the peaks and increasing troughs (depending on region/behavior) during social interaction and drinking behavior suggest some evolution of recruitment.

Unsupported male-female post hoc comparisons in the manuscript have been removed. Focus has been shifted to effects seen in females and effects seen in males.

Baselines used has been added and is a conservative measure. The addition of individual average heatmaps is useful, although there is some concern about the variability of signals observed within a subject, as that information is lacking.

Understand about the 4 h versus 2 h for drinking data.

Dear Reviewers,

Thank you for your helpful suggestions regarding our revised manuscript. We believe we have addressed all concerns in our second revision. We respond to each of these concerns in a point by point response below, with our responses noted in red. All changes in the revised manuscript are noted in blue.

Sincerely,

Thomas Kash

Reviewer #1 (Remarks to the Author):

I thank the authors for thoroughly responding to my comments. The revised manuscript is much improved in my opinion. The addition of the DREADD experiments in figure 8 certainly addresses a major concern with the descriptive nature of the first submission. However, there are still a few areas of concern that should be addressed.

1) I would like to push back slightly on the responses to my comments on the fiber photometry experiments in figures 1 and 6. Choosing 5 seconds still seems very arbitrary considering that the signal does not appear to return to baseline until well after 5 seconds in the case of social interaction, water drinking, and ethanol drinking. This is even more interesting considering the stability of the data prior to the bout now that it is clearer that the Z-score is not calculated within this specific window.

Of equal concern is the acoustic startle, which returns to baseline much quicker than 5 seconds. To the eye looks like there are several differences that are going undetected with the current analyses.

a) Based on how the data are presented in figure 1O, looks to me to be a very large male/female difference in peak response (more than double), but the average over 5 seconds is not different.

b) GRAB-5HT data in 1AA and II- similar peak but male activity stays elevated much longer than female

c) Figure 6 M vs O and Q vs S- male is significant but this is driven by an increase well after the stimulus has ended, whereas the peak z-score is similar in males and females, if anything both are trending towards an increase

d) Figure 6 GG- the DiD response is quite different than the water and again an effect at the time of the stimulus appears to be washed out by a negative bump well after the stimulus.

I fully appreciate the complexity of the analysis and interpretation of the results and wanting to perform uniform analyses across experiments, but in this case, uniformity detracts from potential interesting sex- and DiD-dependent differences. The authors mention in the response (but not the main text) that maximum peak and AUC data are negative across the board, but based on the visual representation of average traces, it would be informative to include these negative data and other behavior specific measurements to address my, and likely other readers, confusion with the data as presented. This could also be very informative for other research attempting to follow up on these interesting results.

While we understand the reviewer's point about wanting to observe traces for the entirety of the social interactions, we do not think this is appropriate due to the inherent variations in the length of time that each animal spends interacting. This is similarly relevant for our drinking experiments, where the animals are choosing when and for how long to engage in the behavior we are measuring. For example, some mice interact for only 5s (the lowest end of the range), while others interact for nearly 30s (the highest end of the range). Therefore, showing an averaged trace that includes all animals and is 30s long would capture animals who are still engaging in social interaction, animals that have stopped and started interacting, and animals that have stopped interacting and begun engaging in another activity in their home cage (such as grooming, rearing, or digging). The 5s window here was thus not arbitrary, but based on the minimum length of time that experimental mice spent engaging in their first social interaction during the test. In addition, we would like to highlight that we were concerned here with the specific moment that the experimental mice interact with the conspecific, as we saw alterations in social recognition but not social interaction. 5s was then chosen as a time window for the minimum length of drinking bouts (criteria) for consistency and the fact that most drinking bouts exceeded this length, but we agree that for startle this large of a window may not have been ideal. We have done a series of additional analyses to attempt to accurately quantify the apparent differences in signals between groups that were not originally reported using shorter time windows and/or curve fitting methods. These include:

- We report a trend for an increased LHb GCaMP peak signal in DiD females during social interaction.
- We report increased BNST GCaMP startle responses in males compared to females (peak height, 0-1s post startle)
- We report increased tau (slower decay) in 5-HT signals in BNST and LHb of males compared to females during startle (0-5s post startle). We did not observe any differences in 5-HT signal decay between DiD and Water groups for either sex or either brain region.
- We report NO differences in neural responses to startle in DiD females compared to water females for GCaMP or GRAB-5HT. However, there was a trend towards an increased GRAB-5HT peak height in the LHb of DiD females during startle.
- We now include bars above the peri-event plots showing the duration of the behaviors/stimuli.

2) I appreciate the authors pointing out the caveats with their work, however I am still concerned with the dorsal vs ventral BNST virus expression. Specifically, the authors state on page 19 line 17-20 that "...in some cases did observe spread to the dBNST". If the spread into dorsal BNST is only in some cases this raises some new concerns. The authors should simply remove the cases where virus spread into the dBNST. This could also clarify some of the difficulties with interpreting the new DREADD data. Alternatively, some sort of control electrophysiology experiment could be performed to convince the reader this spread is not driving the results of the DREADD experiments.

a) I'm similarly concerned that the representative image of fiber placement for BNST GCaMP (Figure 1B) is in the dorsal BNST while the GRAB-5HT image is in the ventral BNST (Figure 1V). The BNST sub-region discrepancies throughout the manuscript should be addressed.

Minor- the authors state that 415 was the isosbestic wavelength used for the experiments but the addition on page 4 line 26 says 405. Please clarify.

- We have corrected the 405 nm error to 415.
- We understand the reviewer's concern about virally targeting the dorsal in addition to the ventral BNST. We would like to point out that the dorsal and ventral BNST are highly interconnected, and because of this even when virus is injected into the ventral BNST directly, it is extremely unlikely that the virus does not also label cells in the dorsal BNST (primarily the most ventral portion of the dorsal BNST). We indeed misspoke when we used the word "some" in reference to animals that had both dorsal and ventral BNST viral expression—in reality, **the vast majority (75%) of mice had expression in both dorsal and ventral BNST**. To illustrate this, we now include schematics of viral infection spread for all of our BNST infections for the 2c deletion, 2c Gi DREADD, and 2c Gq DREADD experiments in Figures S13-S14.
- With regards to fiber photometry fiber placements, we now also include schematics of fiber placements for all of our animals (Figure S12). We would like to point out that even in cases where the fiber tip is in the dorsal BNST, it is in the very lowest part of the dorsal BNST, which places it primarily above ventral BNST neurons that likely provide the bulk of the signal. According to Pisanello et al. (2019), the collection volume for a 0.39 NA, 200 μ m diameter optic fiber is 10^6 microns³, and 90% of signal is collected from this volume extending ~300 microns from the fiber tip. Thus, even our fibers located in the dBNST would have collected a substantial amount of vBNST signal (see figure 1 below). Moreover, when we exclude dBNST placements, our overall main findings are unchanged (see figure 2 below, though groups do become underpowered with removal of dBNST animals).

Figure 1:

Figure 2:

Reviewer #2 (Remarks to the Author):

The addition of mechanistic data (the chemogenetic inhibition experiments) improves the story and excitement for this manuscript. The novel take home from this work is that the LHb population is important for disturbances to social behavior following binge-like drinking in female mice. However, this appears not to be due to 5HT2c, but instead in relation to overactivation of this population. Overall, the paper still reads a bit descriptive and slightly confusing in places, but is improved in flow and now more focused on the main findings. In regards to the new data, they point to a novel role for the LHb5HT2c population in social deficits and increased alcohol consumption in females, but not males, while chemogenetic inhibition of the BNST 5HT2c population normalized startle behaviors.

My apologies for not understanding the drinking bout duration criteria. For photometry traces, a bar above the graph indicating ongoing behavior would be useful for many reasons, first to avoid confusion about whether animals are behaving differently, but also in regards to the dynamics that evolve across the period. While the behaviors are unlikely to be exactly the same, the peaks and increasing troughs (depending on region/behavior) during social interaction and drinking behavior suggest some evolution of recruitment.

Unsupported male-female post hoc comparisons in the manuscript have been removed. Focus has been shifted to effects seen in females and effects seen in males.

Baselines used has been added and is a conservative measure. The addition of individual average heatmaps is useful, although there is some concern about the variability of signals observed within a subject, as that information is lacking.

Understand about the 4 h versus 2 h for drinking data.

We appreciate the reviewer's comments. We have added bars above the photometry traces as per your suggestion. While showing each individual trial for each test for each animal is not entirely feasible due to journal figure constraints (we now have 16 supplemental figures), we have compared first vs. last trials for all behaviors tested (GCaMP and GRAB-5HT). We now include the only non-negative findings from this analysis as an additional supplemental figure (Figure S7). While there is no difference in LHb or BNST first vs last trial response to startle in naive males, when separating out first vs. last startle trial for DiD vs. Water males we observe that the increase in DiD animals was driven by early trials for the LHb and by late trials for the BNST. This could suggest that the BNST and LHb are both important for startle potentiation, but that alcohol mediates initial startle potentiation via the LHb and sustained startle potentiation via the BNST. Together, we believe this additional data addresses the reviewer's concerns regarding how signals adapt across trials during testing.

REVIEWERS' COMMENTS

Reviewer #1 (Remarks to the Author):

The authors have adequately addressed my remaining concerns. The manuscript is much improved.

Reviewer #2 (Remarks to the Author):

The reviewers have addressed my concerns